



# VOLCANO₃ - A Miniaturized Chemiluminescence Ozone Monitor for Drone-Based Measurements in Volcanic Plumes

Maja Rüth[1], Nicole Bobrowski[1,2], Ellen Bräutigam[1], Alexander Nies[3], Jonas Kuhn[4], Thorsten Hoffmann[5], Niklas Karbach[5], Bastien Geil[5], Ralph Kleinschek[1], Stefan Schmitt[6], Ulrich Platt[1]

[1] Institute of Environmental Physics, University of Heidelberg, Heidelberg, Germany

[2] Istituto Nazionale di Geofisica e Vulcanologia, Osservatorio Etneo, Catania, Italy

[3] Laboratoire de Physique et Chimie de l`Environnement et de l`Espace, CNRS/University Orléans, Orléans, France

[4] Department of Atmospheric and Oceanic Sciences, University of California, Los Angeles, CA, USA

[5] Institute of Inorganic and Analytical Chemistry, Johannes Gutenberg-University, Mainz, Germany

[6] Airyx GmbH, Hans-Bunte-Str. 4, 69123 Heidelberg, Germany

*Correspondence to*: Nicole Bobrowski (nicole.bobrowski@ingv.it)

**Abstract.** High levels of bromine monoxide (BrO) observed in volcanic plumes indicate significant catalytic destruction of tropospheric ozone ($O_3$) at local to regional scales. The underlying chemical mechanisms are still poorly understood and the

quantification of $O_3$ concentrations and their distribution in volcanic plumes remain a major challenge. Common atmospheric $O_3$ measurement techniques (UV absorption and electrochemical sensors) suffer from strong interferences, especially from sulphur dioxide ($SO_2$), which is low in the atmospheric background but a main constituent of volcanic plumes (ppmv levels). This problem can be circumvented by using chemiluminescence (CL) $O_3$ monitors, which have no known interference with $SO_2$ and other trace gases commonly found in volcanic plumes. However, volcanic plume measurements with modern CL $O_3$

monitors are impractical because they are heavy and bulky. Here we report on the development and application of a lightweight version of a CL $O_3$ instrument (l.5 kg, shoebox size) that can be mounted to a commercially available drone. Besides measurements of vertical $O_3$ profiles over several hundred metres, we present drone-based CL $O_3$ measurements in the volcanic plume of Mount Etna in Italy. Within 3 km of the emitting craters we measured an anti-correlation between $SO_2$ and $O_3$ concentrations, corresponding to ozone reductions by up to 60 % in the volcanic plume with respect to the surrounding

atmosphere.

## 1 Introduction

Earth's stratospheric ozone ($O_3$) layer absorbs the short-wavelength part of the solar ultraviolet radiation, enabling life as we know it. Besides its prominent role and abundance in the stratosphere, smaller amounts of $O_3$ in the troposphere play an important role in the oxidation chemistry. Globally, tropospheric $O_3$ is the most important precursor of hydroxyl radicals (OH),





which drive chemical conversion and removal of many pollutants and greenhouse gases. At the same time, tropospheric $O_3$ itself acts as a greenhouse gas, contributing to global warming. While today surface $O_3$ concentrations are routinely monitored (e.g. Cooper et al., 2014), measurements of the vertical profile with high spatial and temporal resolution are rare, yet highly desirable. Consequently, there remain major gaps in our understanding of tropospheric $O_3$ sources and sinks, interaction of transport processes with $O_3$ chemistry, and the detailed impact of $O_3$ on the atmospheric composition.

For a long time, $O_3$ has been measured with chemiluminescence (CL) techniques. In fact, nitric oxide and ethylene CL measurements of $O_3$ are still the standard method in the United States (USEPA, 2023) and are considered the most reliable $O_3$ measurement methods (e.g. Long et al., 2014, Long et al 2021). Nevertheless, nowadays and since many decades, most $O_3$ measurements use short-path UV-absorption spectroscopy (e.g. Dunlea et al., 2006, Williams et al., 2006), which is much easier to implement, cost-effective, and, in most environments, similarly accurate. Such devices measure $O_3$ concentrations by

detecting the attenuation of radiation around 254 nm. Typically, absorption measurement paths are a few tens of cm and the light intensity without $O_3$ absorption needs to be monitored (e.g. once a minute) by passing the air through an $O_3$-remover ("$O_3$ scrubber"). But volcanic and biomass burning plumes are examples of tropospheric environments, where the applicability of UV absorption $O_3$ monitors is questionable, because of interfering UV absorption due to $SO_2$, volatile organic carbon species, fine aerosol, and mercury vapour. For example, looking at the absorption cross-sections of $O_3$ and $SO_2$, it can be recognised

that the sensitivity to $SO_2$ is about a factor of 100 lower than that to $O_3$ by considering the wavelength which the $O_3$ maximum absorption, where many UV-monitors operate ($\approx$254 nm). Under most atmospheric conditions these interferences (especially due to $SO_2$) are negligible (Kleindienst et al., 1993; Williams et al., 2006) since ambient $SO_2$ levels are typically comparable to or lower than $O_3$ levels. Therefore, UV monitors can reliably measure $O_3$ at most remote, urban, and industrial locations. However, when probing volcanic emissions, $SO_2$ mixing ratios may reach values up to several ten ppmv and therefore can

exceed $O_3$ mixing ratios by factors of 1000 or more (usual background $O_3$ mixing ratios are several ten ppbv). Consequently, $SO_2$ typically dominates UV absorption in volcanic plumes and prohibits an accurate quantification of the $O_3$ UV absorption signal (Kleindienst et al., l993; Leston et al., 2005; Williams et al., 2006). The correction of the data with simultaneously measured $SO_2$ (Kelly et al., 2013) or the application of selective $SO_2$ scrubbers (Surl et al., 20l5; Vance et al., 20l0), however, are difficult and – at best - introduce significant additional uncertainty.

A similar problem can occur in biomass burning plumes, typically containing high levels of hydrocarbons and particles, which can also produce false $O_3$ signals in the UV (Cavanagh and Verkouteren, 2001, Kleindienst et al. 1993).

Other $O_3$ measurement techniques, which are in use since decades, for instance electrochemical sensors for balloon-borne profile measurements of the atmospheric background (Witte et al., 2017) show strong cross interferences to higher amounts of nitrogen dioxide and $SO_2$ (Schenkel et al., 1982), which also makes them unreliable in near-source plume environments.


The VOLCANO$_3$ instrument implements the 'traditional' CL $O_3$ measurement technique in a compact, robust, and light-weight setup. It can be mounted to a commercial drone and thereby provides accurate and interference-free $O_3$ measurements with meter-scale spatial resolution in all tropospheric environments. In this work we focus on the application of drone-based CL $O_3$





measurements in volcanic plumes with the goal of improving the understanding of volcanic plumes and their impact on the
atmospheric composition.

## 1.1 Volcanic gases and ozone

Volcanoes are a key component of Earth's element cycles and have an impact on their local environment, particularly the surrounding atmosphere. However, their impact can also be regional and global in scale when emissions increase during eruptions or when quiescently degassing for a long time (e.g. Marti and Ernst, 2005, von Glasow et al., 2009).

The primary volcanic gas emissions, i.e. water vapour, carbon dioxide ($CO_2$), sulfur species ($SO_2$, $H_2S$), and hydrogen halides (HCl, HF, HBr, HI) mix and interact with the surrounding atmosphere, creating a unique atmospheric environment (e.g. Carroll and Holloway, 1994, Kuhn et al., 2022). For example, high amounts of secondary reactive halogen species, especially bromine monoxide (BrO), have been detected in volcanic plumes (e.g. Bobrowski et al., 2003, 2007; Kern et al., 2009; von Glasow, 2010; Gliß et al., 2015; General et al., 2015). This indicates heterogeneous photochemical reaction cycles (referred to as bromine

explosion, see Platt and Janssen, 1995, Wennberg, 1999), involving volcanic halogen halides, aerosol particles, and atmospheric oxidants (mainly $O_3$, see Bobrowski et al., 2007; Kern et al., 2009; Jourdain et al., 2016). The reactive cycles include the catalytic destruction of $O_3$, which has led to the widespread assumption that $O_3$ levels in volcanic plumes are depleted with respect to the atmospheric background.

The conclusion that the observed amounts of reactive halogens in volcanic plumes lead to depleted $O_3$ levels is, however, by

no means trivial. Field studies (using CL as well as short-path UV absorption instruments) have shown varying degrees of $O_3$ depletion across different volcanoes, in some cases up to 90% $O_3$ loss compared to ambient levels were reported (e.g. at Mount St. Helens, USA, see Hobbs et al., 1982). In other cases, no $O_3$ depletion was found (e.g. at Kilauea, Hawaii, USA, see Roberts, 2018) which was explained by low concentrations of halogens. Elementary calculations assuming a constant influx of $O_3$, a basic turbulent mixing scheme and the BrO self-reaction as a rate determining step for the $O_3$ destruction in volcanic plumes,

suggest that the influx should largely compensate $O_3$ destruction. Therefore, these simple calculations predict negligible (<1%) $O_3$ destruction in volcanic plumes (Rueth 2023). Conversely, model studies with more evolved multiphase atmospheric chemistry mechanisms predict significant destruction of $O_3$ in volcanic plumes (e.g. Surl et al. 2021, Nies et al. 2025, see Supplementary Material Fig. S4) in accordance with some observations (Surl et al. 2015).

Measuring $O_3$ levels in volcanic plumes is challenging as pointed out above. The aim of this study is to provide a technique

for reliable $O_3$ measurements in volcanic plumes. Building upon previous studies, this work focuses on employing gas-phase chemiluminescence (CL)-based $O_3$ monitors for volcanic plume measurements (Hobbs et al., 1982; Vance et al., 2010; Carn et al., 2011).

After a short summary of the CL $O_3$ monitor principle (Sect. 2), we introduce our small and lightweight CL $O_3$ monitor, VOLCANO$_3$ (Sect. 3). In Sect. 4, we present field measurements including planetary boundary profiles and $O_3$ measurements

in the volcanic plume of Mt. Etna, Italy, which show significant $O_3$ depletion compared with the ambient atmosphere. We further discuss future technical developments (Sect. 5) and applications (Sect. 6) of VOLCANO$_3$.





## 2 The principle of CL $O_3$-Monitors

The principle of operation relies on the generation of chemiluminescent species through reactions involving $O_3$. By measuring the emitted photon flux, it is possible to infer the concentration of $O_3$. There are various CL reactions involving $O_3$ that are

employed in CL $O_3$ monitors, with $C_2H_4$ being the most commonly employed reactant and which is used also in this study. Nederbragt et al. (l965) were the first to make use of this chemiluminescent reaction to determine $O_3$ near an accelerator and Warren and Babcock (l970) then described the construction and calibration of such a monitor.

The reaction of $O_3$ with $C_2H_4$ produces various products (detailed in e.g. Kleindienst et al., 1993, Rüth, 2023), including electronically excited species that emit photons upon de-excitation. The number of emitted photons, directly measured by a

photomultiplier tube (PMT), is proportional to the $O_3$ concentration and thus to the $O_3$ volume mixing ratio $X_{V,O_3}$. In addition to being proportional to the $O_3$ mixing ratio, the number of photons emitted per second is influenced by several other parameters, including the quantum yield of the reaction, the ambient temperature and pressure, as well as the concentration of $C_2H_4$, regulated through the $C_2H_4$ flow ($f_{C_2H_4}$) in ml/s (or $10^{-6}$ m³/s). In order to determine the $O_3$ volume mixing ratio ($X_{V,O_3}$), the measured photomultiplier tube (PMT) signal has to be converted according to the theoretical description and calibrated

experimentally using an $O_3$ generator,

$$X_{V,O_3} = \frac{\gamma k_b}{Q} c_{con}\left(f_{C_2H_4}, p, T\right) \qquad (1)$$

$$c_{con}\left(f_{C_2H_4}, p, T\right) = \frac{T}{p(f_{tot} - f_{C_2H_4})\left[1 - exp\left(-k_1(T)\frac{f_{C_2H_4} p V_{cell}}{f_{tot} k_b T}\right)\right]} \qquad (2)$$

where $\gamma$ denotes the number of photons generated per second, $Q$ the detector quantum efficiency of the PMT, $k_1(T)$ denotes the reaction rate constant of $O_3$ with $C_2H_4$ in cm³/molec, p is the ambient pressure, $T$ the ambient temperature, $V_{cell}$ the

measurement cell volume (in this study 20 ml), $f_{tot}$ the flow rate of the pump in ml/s (or $10^{-6}$ m³/s) with $V_{cell}/f_{tot}$ the residence time in the cell, $k_b$ the Boltzmann constant and $c_{con}\left(f_{C_2H_4}, p.T\right)$ is the "conversion factor" given in $10^{-6}$ Ks/J (or Ks/Pa·ml = Ks/(Pa·$10^{-6}$ m³)).

Note that the quantum efficiency $Q$ also includes the probability of a CL photon to actually reach the photocathode of the PM. We estimate this probability at about 15% for a reflectivity of the reaction chamber walls of $R_W = 0.7$. For an in-depth

description of the theoretical basis see (Rüth, 2023, Bräutigam, 2022).

## 3 A compact CL ozone monitor

### 3.1 Configuration

In Fig. 1 simplified schematic drawing of the monitor VOLCANO$_3$ is shown (Bräutigam, 2022). It has the dimensions 38 x 20

x 11 cm and weighs around 1,5 kg. VOLCANO$_3$ consist of the following components:1) a reaction chamber, 2) a pump, 3) an ethylene minican 4) a photomultiplier module, 5) a circuit board and a Raspberry Pi computer and 6) a lithium-polymer battery.



Ambient air enters the instrument through an aerosol filter (: Schematic drawing of the CL $O_3$ monitor setup (without the electronics). Figure taken from Bräutigam (2022). (A)) and is then directed trough a black Teflon hose (two windings to suppress ambient light entering the measurement cell trough the hose) into the measurement cell (aluminium with a volume $V$

= 20 mL, Figure 1, (B)), where it is mixed with $C_2H_4$. The measurement cell is fixed to the PMT photo cathode (Fig. Figure 1, (C)) enabling the measurement of the photons emitted by the air + ethylene mixture. The PMT (Hl0493-00l from Hamamatsu Photonics GmbH) is the central part of the instrument and main reason for the weight and size reduction due to its internal high voltage supply and preamplifier electronics. In contrast to the most commonly applied CL-monitors, the PMT is operated without temperature stabilisation (rather the temperature effect is compensated during signal evaluation, see Sect. 3.2), thus

significantly reducing the power consumption of the monitor to only 3 W. Ethylene is supplied from a minican ($V_{bottle}$ = 1 L, maximum overpressure $p_{bottle}$ = 12 bar (Figure 1, (D)). The $C_2H_4$ flow into the measurement cell is regulated via a pressure regulator and a capillary.

The gas mixture leaves the instrument through the airflow generated by the pump (model G 6/0l-K-LC). Ambient temperature and pressure, as well as the temperature at the PMT are also monitored in order to convert the measured signal to the $O_3$ mixing

ratio. All relevant data (date, time, temperature, ambient pressure, output voltage of the PMT, pressure of the minican) is recorded by the microprocessor on a USB drive and also shown on a small display to ensure operational functionality of the device during the measurement.

## 3.2 CL-Monitor Characterization

### 3.2.1. PMT Dark current

Since the PMT used is not temperature stabilised, the dark current (measurable signal if no photons hit the photosensitive area) is a temperature-dependent variable for which a correction is required. The temperature dependence of the dark current compiled from several measurements is shown in Figure 2.

In order to subtract the dark current as a function of temperature from the data the Richardson function (see Eqn. 3) is fitted:

$$S_0(T) = a_1 \cdot T^2 \cdot exp\left(\frac{b_1}{T}\right) + c_1 \tag{3}$$

The parameters $a_1$, $b_1$, $c_1$ are determined in regular intervals. Typical values are $a_1 = (9.9\pm0.3)\cdot10^{12}$ mv/$K^2$, $b_1 = (-1.2\pm0.001)\cdot10^4$ K, and $c_1 = (0.55\pm0.003)$ mV.

The uncertainty arising from the dark current correction is determined by the mean deviation of the measurement to the fit $S_{0,mean}$, leading to a propagation into the uncertainty of the $O_3$ determination of only 1 ppb. Temperature is the dominant driver of the dark current signal, minor signal deviations are additionally corrected by an easy pragmatical solution, subtracting the

$S_{0,mean}$ from each measurement. Then the dark current corrected signal $S_{corr}$ is calculated as:

$$S_{corr} = S - S_0(T) - S_{0,mean} \tag{4}$$

Where S is the measured signal, $S_0(T)$ the temperature dependent dark current as derived by Eq. 3, and $S_{0, mean}$ the mean fit residuals from the dark current temperature correction given in mV, because the output signal of the PMT is measured in mV.



### 3.2.2 Calibration

The CL method is not a direct measurement technique, therefore, in order to calculate the $O_3$ mixing ratio from the measured PMT-signal an experimental calibration is needed, based on the theoretical considerations in Sect. 2. (Eq. 1 and 2):

$$\gamma = a_{cal} \cdot S_{corr} \tag{5}$$

Combining the above equations (2 and 5) we obtain an equation for the $O_3$ mixing ratio $X_{V,O_3}$:

$$X_{V,O_3} = a_{cal} \cdot c_{con}(f_{C_2H_4}, p, T) \cdot S_{corr} \tag{6}$$

where $c_{con}$ is the theoretically determined conversion factor (see Sect. 2.) and $a_{cal}$ the calibration constant. Here, no calibration offset is necessary, since any potential offset in the data is taken care of by applying the dark current correction.

The CL $O_3$ monitor is calibrated using an $O_3$ generator, primarily the Ozone Calibration Source Model 306 by 2B Technologies. It is a portable $O_3$ generator and can provide $O_3$ in the range of 0 to 1000 ppb. Additionally, the $O_3$ generator ANYSCO type

SYCOS KT-$O_3$/$SO_2$, which can provide 0 and 150 ppb of $O_3$, was used.

To calibrate the monitor several calibration measurements with varying $O_3$ mixing ratios in different sequences are made. For the periods of constant $O_3$, the converted signals are averaged and plotted against the sampled $O_3$ mixing ratios as shown in Figure 3. A linear fit is performed; its fit parameters are then used for the calculation of the $O_3$ concentrations.

### 3.2.3 Detection limit and measurement uncertainty

Both, the detection limit and the measurement uncertainty are crucial characteristics of any instrument. Following the definition of Gold (2019) the detection limit (also referred to as limit of detection (LoD)) is the 'minimum single result which, with a stated probability, can be distinguished from a suitable blank value' (Gold, 2019, p. 399). To determine the uncertainty of the mixing ratio of $O_3$, error propagation is applied to Eq. 6 for $X_{V,O_3}$:

$$\Delta X_{V,O_3} = \left[\left(\frac{\partial X_{V,O_3}}{\partial S}\Delta S\right)^2 + \left(\frac{\partial X_{V,O_3}}{\partial S_0}\Delta S_0\right)^2 + \left(\frac{\partial X_{V,O_3}}{\partial c_{con}}\Delta c_{con}\right)^2 + \left(\frac{\partial X_{V,O_3}}{\partial a_{cal}}\Delta a_{cal}\right)^2\right]^{\frac{1}{2}} \tag{7}$$

Signal uncertainty, $\Delta S$, is deduced from the standard deviation $\sigma_s$ of constant $O_3$ periods, represented by:

$$\Delta S = \sigma_s = a \cdot ln(b \cdot S + c) \tag{8}$$

$\Delta S_0$ is the uncertainty from temperature-corrected dark current, approximated as the mean deviation from the measurement to the fit, around 0.5 mV. Calibration uncertainty is quantified by $\Delta a_{cal} = 24$ (hPa·ml/(mV·K))

The uncertainty from signal conversion is estimated from the mean deviation of experimental data to theoretical description,

about 4%. For a representative signal of $S = 20$ mV, corresponding to $O_3$ mixing ratio of 40 ppb, the uncertainty analysis is:

$$\Delta X_{V,O_3} = 0.03 \cdot X_{O_3} \tag{9}$$

The detection limit, assuming $\sigma_{S,0} \approx 0.4$ mV, is approximated as:

$$X_{V,O_3,0} \approx 1.13 \text{ ppb} \tag{10}$$



### 3.2.4 Response time

In order to determine the response time of the monitor experimentally, step changes in the $O_3$ mixing ratios are of interest. To circumvent the generator's response time, one can produce a consistent $O_3$ mixing ratio using the generator, but without attaching the hose to the monitor. Once a stable mixing ratio is achieved, the hose can be swiftly connected to the monitor. The step change recorded in this manner should reflect the response time of the $O_3$ monitor.

To these increasing and decreasing step changes exponential increases and decreases are fitted, respectively. For the calculation
of the response time increases and decreases are treated equally and only fits with a $R^2$ larger than 0.9 are used. The experimental response time changes of the monitor, determined from 25 step changes fulfilling the above requirements, results to $\tau_{\exp,1/e} = 2.00 \pm 0.04$ s.

### 4 Field Measurements

### 4.1 Vertical atmospheric profile

Vertical atmospheric $O_3$ profiles in the planetary boundary layer (PBL) with a high spatial resolution are still surprisingly rare. Some $O_3$ profiles were taken in the frame of this study using an UAV (Matrice300 RTK) as a vehicle to transport the VOLCANO$_3$ monitor to different heights of the PBL. One example of our measurement results is given in Fig. 4. The measurement was carried out at the southern flank of Etna, at Piano Vettore, Italy on 16th of June 2023 and shows the increase of $O_3$ mixing ratios from about 50 ppb to nearly 100 ppb for an elevation change from about 1700 m.a.s.l. to about 2500 m.a.s.l.

### 4.2 Measurements in the volcanic plume of Mt. Etna

As mentioned in the introduction, the CL technique has particularly unique advantages when applied to $O_3$ determination in volcanic plumes. Therefore, we performed a first measurement campaign at the volcano Mt Etna to investigate the $O_3$ distribution of its plume.

### 4.2.1 Measurement Site and conditions

Mt. Etna, located at the eastern coast of Sicily, is one of the most active volcanoes globally and is characterized by significant continuous gas emissions. Geological evidence suggests volcanic activity since approximately 0.6 million years. The summit area of Etna, reaching to about 3350m above sea level during the campaign, is undergoing significant morphological changes over time, currently hosting four active summit craters: Bocca Nuova (BN), Voragine (VOR), Southeast crater (SEC), and Northeast crater (NEC).

Our field measurement campaign was conducted from June 5th to l8th, 2023. During this period, Etna exhibited continuous outgassing primarily at Bocca Nuova and Southeast crater, with moderate $SO_2$ fluxes and consistently high $CO_2$ levels (INGV National Institute of Geophysics and Volcanology, 2023a, b).





### 4.2.2 Instrumentation and data evaluation

The measurement campaign employed the following instruments:

- CL $O_3$ monitor "VOLCANO$_3$"
- $SO_2$/$CO_2$ sensor "little-RAVEN" (Karbach et al., 2022)
- Drone "Matrice 300 RTK", DJI, https://enterprise.dji.com/de/matrice-300/specs

The little-RAVEN sensor system, designed around an ESP microcontroller, manages various sensors to determine $SO_2$, $CO_2$, temperature, humidity, pressure, and GPS location. It logs data onto internal memory and also transmits it to a ground station,

allowing real-time localisation of the plume and confirming plume gas measurements. The Matrice 300 RTK drone, with a maximum payload of 2.7 kg, was utilized for carrying the VOLCANO$_3$ CL $O_3$ monitor and the little-RAVEN system. The total equipment weight for the measurements was approximately 15 kg, including the instruments, the drone, and the ground station consisting of a notebook, a tripod and an antenna.

### 4.2.3 Volcanic plume measurements and results

During the campaign, a total of four drone flights through the volcanic plume were conducted. The flight paths of all four plume measurements are shown in Figure 5 and the main findings of the four plume measurements are summarised in Tab. l. Flight 1) On June l3th, the VOLCANO$_3$ mounted on the drone sampled the BN plume (black flight path in Fig. Figure 5). The drone started at the Barbagallo craters (the upper pyroclastic cone from the 2002 eruption), south westerly of the SEC (see Fig. S1 in the supplementary). Fig. S1 displays the data obtained from this flight. The meteorological conditions were sunny with

some clouds and with wind blowing mainly from North with low wind speeds. Due to the low wind speeds the plume rose before drifting to the south.

On June l8th, three more flights were carried out:

Flight 1) The first flight on that day started south-southeast of the Barbagallo craters, north-northwest of the Cisternazza (a subsidence crater from 1792) and the pyroclastic cone form the 2001 eruption and navigated through the SEC plume, the

measurement is depicted in Fig. 6 and the flight path shown in Figure 5 in blue color. $O_3$ and $SO_2$ exhibit a strong anticorrelation with a Pearson correlation coefficient of -0.64 and an $R^2$ value of 0.41 (see Tab. 1).

Flight 2) The subsequent flight first sampled the SEC plume before measuring inside the BN plume. This measurement is shown in Fig. S2, the flight path can be seen in red on Fig. 5.

Flight 3) The final flight on the 18[th] of June 2023 focused solely on sampling the BN plume, which can be seen in Fig. S3. The

flight path is indicated in orange on Fig. 5.

These two latter flights started from the northern rim of the Barbagallo crater. The weather was sunny, and only little wind, mainly from the north, this meant that the plume basically ascended and was not pushed down.

In all of these measurements an anti-correlation of $O_3$ and $SO_2$ levels is visible. The ambient $O_3$ fluctuations are in the range of 5 ppb, whereas within the plume variations of up to 60 ppb are observed. Unfortunately, the $SO_2$ sensor is only able to



measure $SO_2$ mixing ratios of up to 16 ppm, leading to higher values being cut off. This obscures part of the correlation. In particular, during the third measurement on the 18th of June 2023, the $SO_2$ sensor frequently reaches its maximum value within the plume, at 16 ppm $SO_2$. Nevertheless, a clear anti-correlation of $SO_2$ and $O_3$ can be observed for all measurement flights, characterized by a $R^2$ of 0.04 – 0.41.

## 5 Future Developments

Based on the first miniaturised VOLCANO$_3$ CL $O_3$ monitor prototype described here a number of improvements of our CL $O_3$ monitor are clearly possible. Our field measurements suggest that the instrument would profit from further size and weight reduction to enhance aerodynamic properties and flight stability of the carrying drone.

The largest components of the instrument are the PMT, the Teflon aerosol filter and the $C_2H_4$ minican. Using a smaller aerosol filter would reduce weight and size significantly. The minican itself is voluminous but quite lightweight, although the minican

adapter is particularly heavy. A custom-made adapter might solve this problem.

Exploring other gases (or liquids with high vapour pressure) than $C_2H_4$ to induce chemiluminescence might lead to a better photon yield of the luminescence and may also present an opportunity to further reduce the size of the instrument. For example, trimethylethylene and tetramethylethylene, $(C_6H_{12})$ might be used. Both species are liquid at room temperature and exhibit a by a factor of 50 higher quantum efficiency and thus emission intensity compared to that of the $C_2H_4$-$O_3$ reaction (Pitts Jr et

al., 1971). Storing the reactant in the liquid phase would significantly reduce the volume of the monitor without reducing the amount of reactant available. With the vapour pressure of 185 mbar (ChemSpider, 2023), $C_6H_{12}$ is available with a mixing ratio of $185/1000 \approx 18\%$. If the $C_6H_{12}$ flow is set to around 10% of the total flow, $C_6H_{12}$ could also be supplied to the measurement cell with a mixing ratio of $\approx 2\%$ (which is the mean mixing ratio of $C_2H_4$ in the current setup).

As mentioned above, the PMT is a rather bulky part of the instrument, in principle it could be replace by one or several

avalanche photodiodes (also known as silicon photomultipliers), which would save space and weight.

Furthermore, the fraction of CL-photons reaching the detector could be enhanced by lining the interior surface of the present aluminium fluorescence cell ($R_W \approx 0.7$) with material of higher reflectivity, for instance with "spectralon-type" material (e.g. ODM98 from Gigahertz Optik GmbH), ($R_W > 0.96$-$0.99$), which would enhance the signal by at least a factor of 2, for details see Bräutigam (2022).

Although the current control unit, based on a Raspberry Pi is not very large it could still be replaced by a smaller microcomputer unit like an ESP for instance. To summarize, there is still room for improvement, but the current instrument is already working excellently.



**6 Discussion and Conclusion**

Our newly developed lightweight VOLCANO$_3$ CL O$_3$ instrument marks a significant advancement in O$_3$ monitoring

technology. Weighing only 1,5 kg and with substantially smaller dimensions compared to commercially available CL-O$_3$ monitors (minimum weight of about 15 kg, which is an order of magnitude heavier and a volume more than five times larger volume as the monitor presented here) (e.g. https://www.teledyne-api.com/en-us/Products_/Documents/Manual/T265%20Manual%20Addendum_07337.pdf). Additionally, a notable reduction in power consumption was achieved, with VOLCANO$_3$ consuming now about 3 W, which is also several times smaller than current

commercially available instruments.

Calibration and ambient measurements in Heidelberg, as well as other measurements in the field, showed the monitor's capability to reliably measure O$_3$. While signal corrections are imperative for deducing the O$_3$ mixing ratio, a detection limit of 1.3 ppb and a measurement accuracy of around 7% for 40 ppb O$_3$ (~ 2.8 ppb) were accomplished, with a response time of $\tau_{exp,1/e} = 2.00 \pm 0.04$ s.

With this miniaturised monitor we successfully detected significant O$_3$ depletions within the volcanic plume ranging up to 60%. While our finding is consistent with some other volcanic plume studies, which also indicate a strong anti-correlation between SO$_2$ and O$_3$, the observed O$_3$ depletion within the volcanic plume presents still an intriguing scientific puzzle. For instance, to fully answer the question on O$_3$ distributions in volcanic plumes and if O$_3$ might be a limiting factor on the bromine transformation in volcanic plumes, more comprehensive measurement campaigns are essential. It would be worthwhile to

study various volcanoes to get a more complete picture. The examination of further high halogen emitting volcanos (additionally to Etna), such as Soufriere Hills and Ambrym as well as rather low halogen emitting volcanos, including the Hawaiian volcanoes such as Kilauea and Mauna Loa, or Copahue in Argentina/Chile would be worth to undertake. Complete plume transects should be conducted in various distances from the volcanic emission source to confirm or disprove our current understanding of the volcanic plume chemistry. Today, our knowledge is mainly based on model studies. Such model studies

show for instance a complete ozone depletion in the centre of halogen rich volcanic plume after a relatively short distance from the emission point (about 10 min downwind, Roberts et al., 2014) but a solid experimental prove of those theoretical consideration is still missing.

This CL-O$_3$ monitor applied on an UAV can also be used in other environments to undertake for instance high spatial resolved O$_3$ measurements in biomass burning plumes, over polar ice fields, salt lakes, in the rain forest or cities, to provide high

resolution maps of the O$_3$ distribution during day and night (e.g., Guimeres et al., 2019). However, the challenges encountered and the intriguing findings regarding O$_3$ depletion in volcanic plumes highlight the potential.

**Acknowledgments**

We thank Christopher Fuchs for his help in the earlier prototype development, Dieter Aletter for his valuable advice and lending us the ozone generator, Heiko Bozem and Peter Hoor for lending us the drone and an ozone generator. Financial



support from MUSUNGU and TeMaS, TPChange, is gratefully acknowledged. We would also like to thank the Istituto Nazionale di Geofisica e Vulcanologia, Italy, grant "Progetto INGV Pianeta Dinamico (MUSUNGU)' grant - code CUP D53J19000170001 - funded by the Italian Ministry MIUR ('Fondo Finalizzato al rilancio degli investimenti delle amministrazioni centrali dello Stato e allo sviluppo del Paese", legge 145/2018).

**Code/Data availability**

The data are available from the author upon request

**Author contribution**

MR and EB are the main developers of the VOLCANO$_3$ instrument with support and contributions from UP, RK, JK and NB.
MR, NB, AN, TH, NK und BG conducted the field campaign at Mt. Etna and provided complementary data. MR evaluated and analyzed the ozone data. All authors contributed to writing of the manuscript.

**Competing interests**

One of the authors (UP) is member of the editorial board of AMT. The authors declare to have no other competing interests.

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




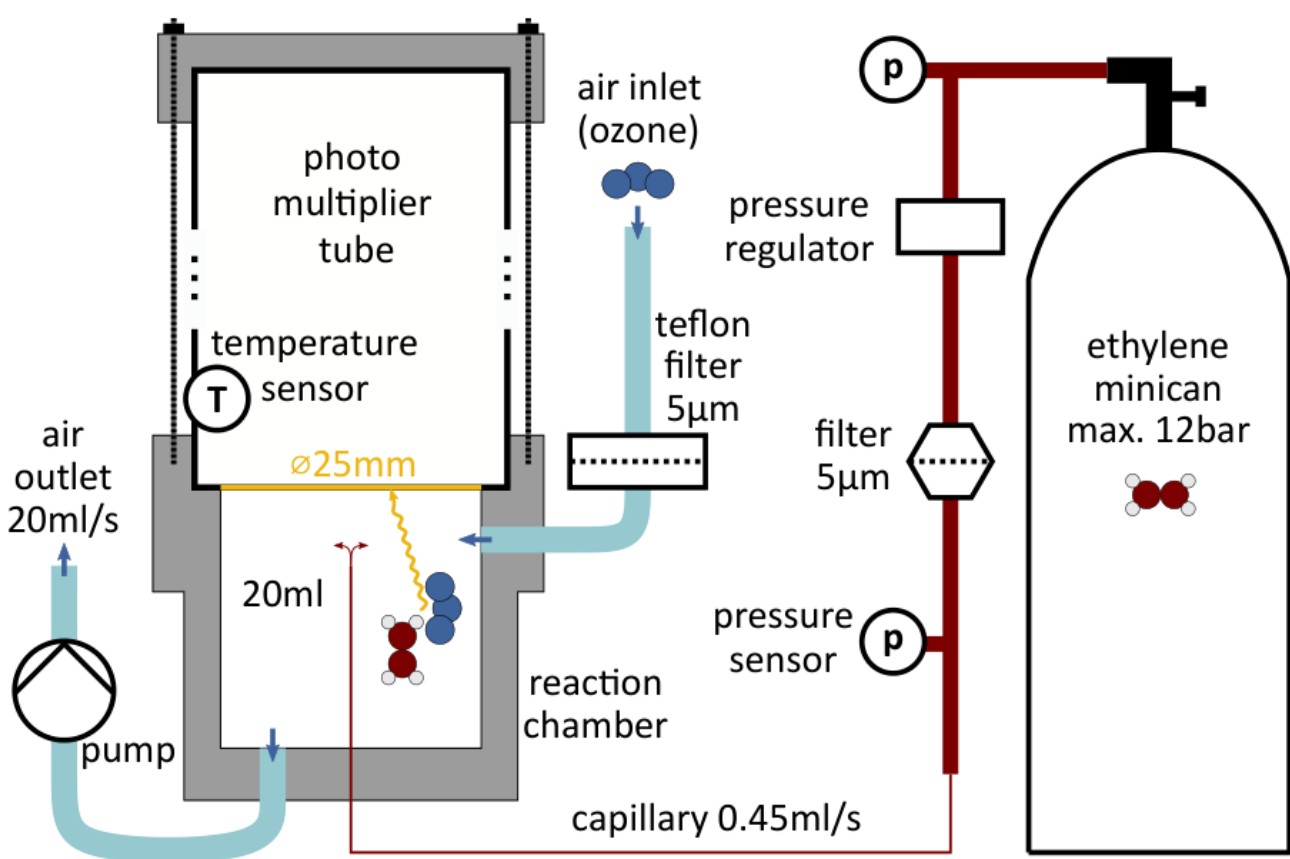

**Figure 1: Schematic drawing of the CL O₃ monitor setup (without the electronics). Figure taken from Bräutigam (2022).**





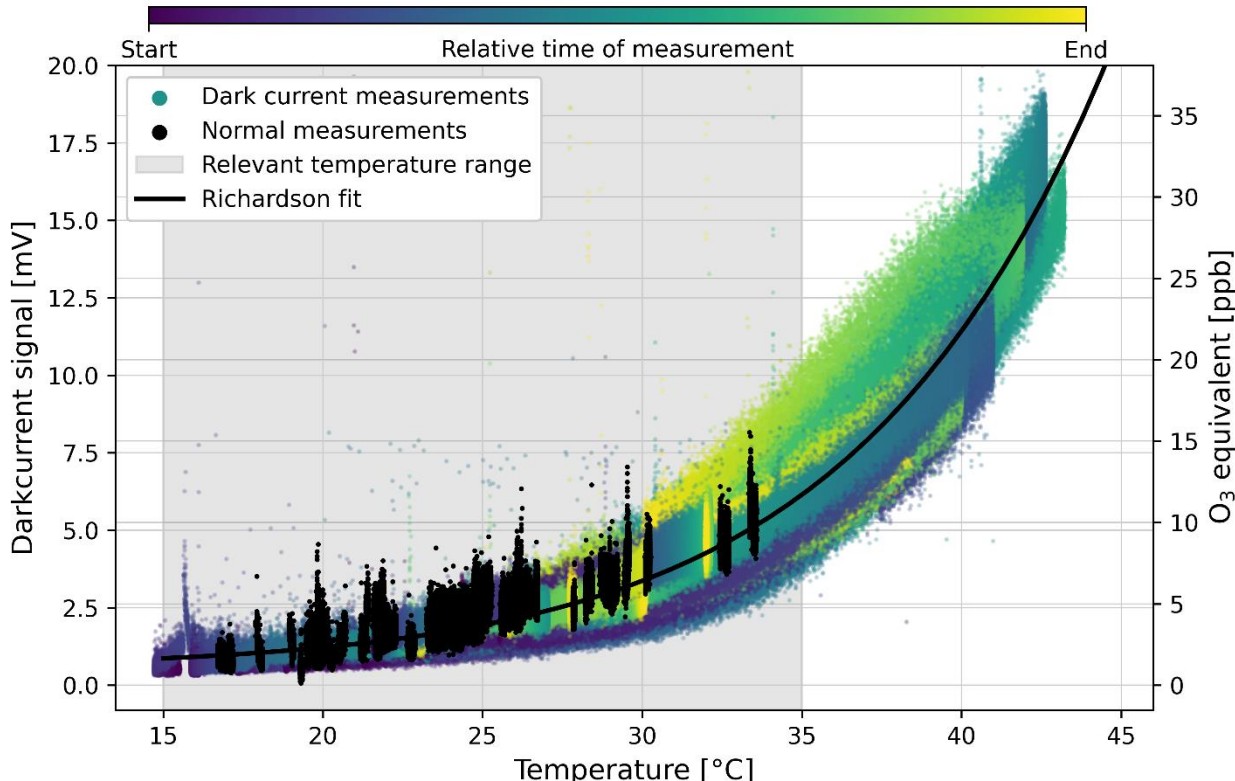


**Figure 2**: **Temperature dependence of the dark current. The colour shade of the datapoints indicate time from blue (start of the measurement) towards yellow (end of the measurement). The black points represent 'normal' O₃ measurements during which also dark current periods were recorded. The data is fitted with a Richardson fit (parameters: a1 = (9.9±0.3)·1012 mv/K2, b1 = (-1.2±0.001)·104 K, c1 = (0.55±0.003) mV) which is then used to correct the temperature dependence of the dark current.**




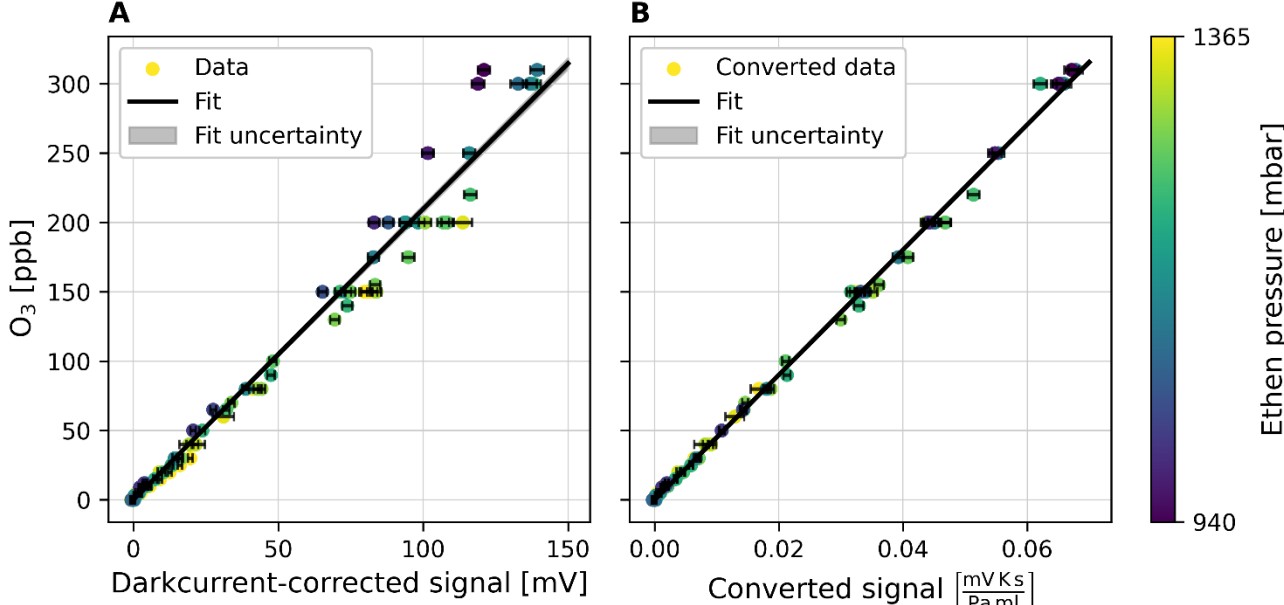

**Figure 3: Calibration plot of the CL O$_3$ monitor. To obtain the data points, the signal is averaged over periods of constant O$_3$ mixing ratios and the uncertainty is given by the standard deviation. The color of the data points indicates the respective C$_2$H$_4$ pressure pC$_2$H$_4$. On the left side, the calibration plot with the dark current corrected signal is shown. Strong deviations from the calibration fit can be seen in the data points. After the conversion, on the right side, the calibration curve fits the data points significantly better. For both cases the calibration fit with the fit parameters as well as root mean squared error (RSME) is shown.**



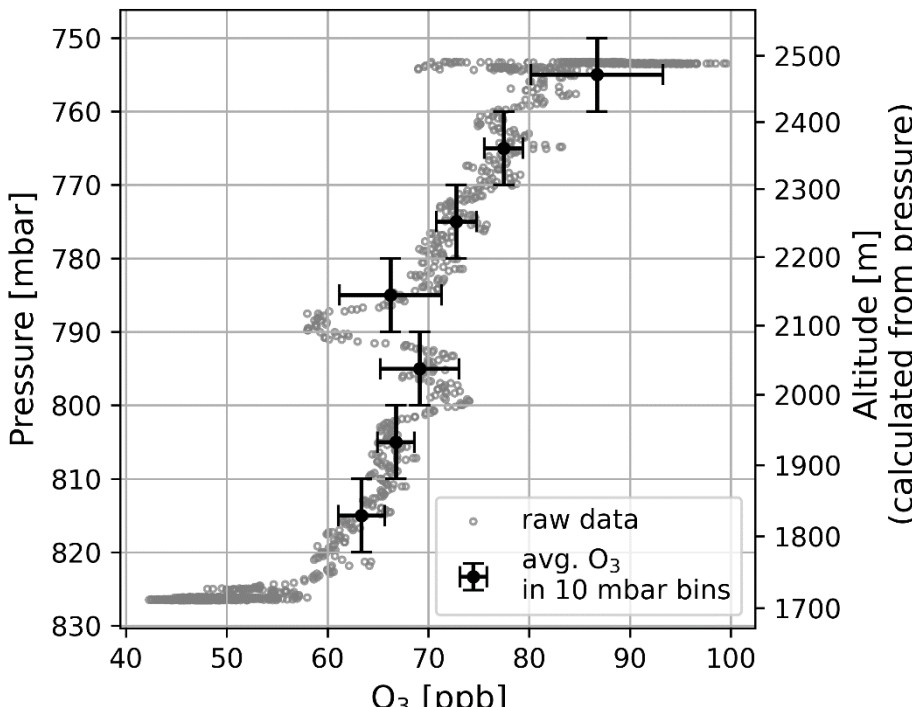

**Figure 4: Vertical O₃ profile for a measurement performed at Piano Vettore on 16.06.2023 during the Mt. Etna field campaign. The data points (and uncertainties) are determined by averaging the O₃ mixing ratio for all data points within a pressure bin of 10 mbar. The O₃ mixing ratio increases with height (increases with decreasing ambient pressure).**







**Figure 5: Map of the summit area of Etna with the flight paths for the four plume measurements. The measurement on 13.06., as**
**well as 18.06, 3, sampled the BN plume. The first flight on 18.06, 1, manoeuvred through the SEC plume and the second flight on**
**18.06, 2, first sampled the SEC plume before measuring the BN plume. Map data: © OpenStreetMap contributors, SRTM | Map**
**display: © Open- TopoMap (CC-BY-SA).**


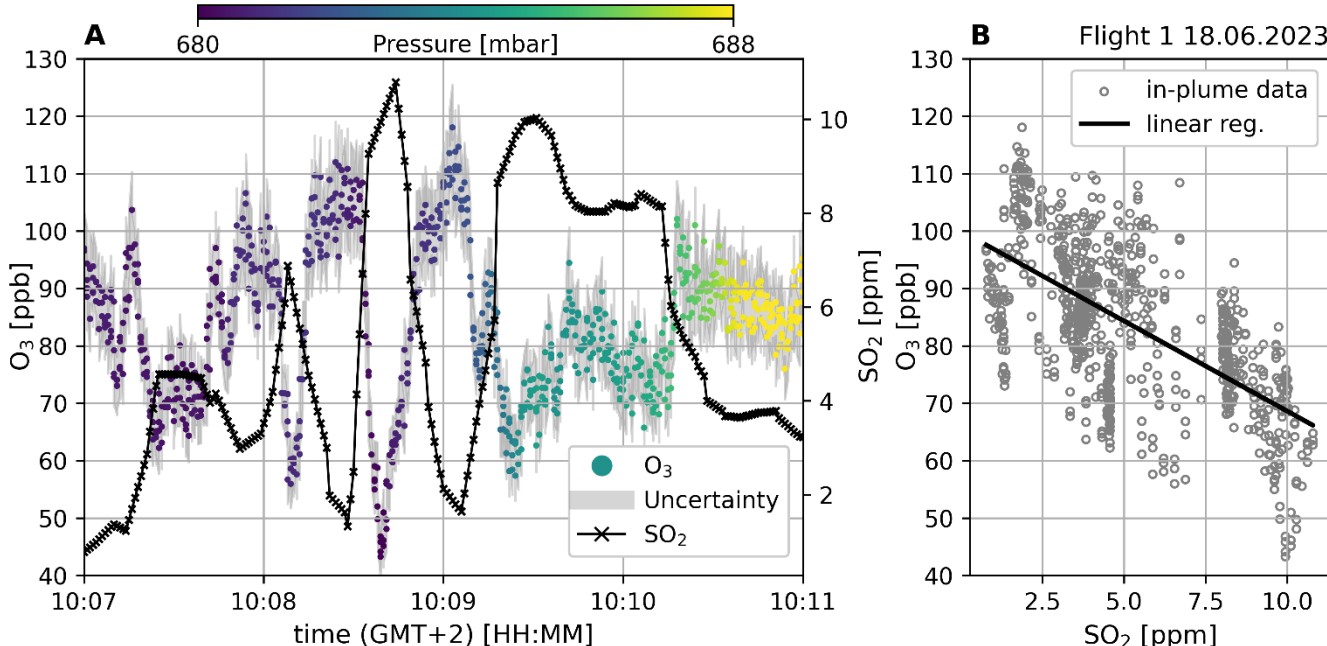

**Figure 6: Plume measurement from 18.06.2023 measuring the SEC plume (see Figure 5 blue line). Panel A shows the time series from the CL instrument and a co-deployed (multi-gas) SO₂ sensor. The colour-coding indicates ambient pressure during sampling**

**and the gray shaded area marks the uncertainty of the CL O₃ measurement. Panel B shows a correlation plot between O₃ and SO₂ for in-plume data points. In-plume datapoints are defined according to the SO2 mixing ratio for values larger than 1.5 ppm and smaller than the saturation value of 16 ppm. The correlation has a Pearson correlation coefficient of -0.64 and an R² of 0.41.**



| Date and Flight | 13.06.2023 | 18.06.2023, Flight 1 | 18.06.2023, Flight 2 | 18.06.2023, Flight 3 |
|---|---|---|---|---|
| **Average distance to crater [km]** | $1.5 \pm 0.5$ | $2.5 \pm 0.5$ | $1.1 \pm 0.5$ | $1.0 \pm 05$ |
| **Mean ambient $O_3$ at ground [ppb]** | $80 \pm 4$ | $66 \pm 4$ | $62 \pm 4$ | $64 \pm 3$ |
| **Mean ambient $O_3$ at plume height [ppb]** | $97 \pm 6$ | $101 \pm 4$ | $111 \pm 4$ | $109 \pm 3$ |
| **Mean $O_3$ in the plume [ppb]** | $93 \pm 5$ | $91 \pm 13$ | $97 \pm 14$ | $91 \pm 13$ |
| **Mean $O_3$ depletion [%]** | 4 | 10 | 13 | 16 |
| **Minimum $O_3$ in plume [ppb]** | 82 | 52 | 60 | 45 |
| **Maximum $O_3$ depletion [%]** | 15 | 49 | 46 | 59 |
| **Mean $SO_2$ [ppm]** | $7.4 \pm 5.2$ | $3.4 \pm 2.5$ | $5.2 \pm 3.9$ | $8.5 \pm 5.4$ |
| **Maximum $SO_2$ [ppm]** | 16* | 9.7 | 16* | 16* |
| **Slope [ppb/ppm]** | $-0.87 \pm 0.07$ | $-3.6 \pm 0.1$ | $-1.9 \pm 0.1$ | $-0.51 \pm 0.07$ |
| **Correlation coefficient** | -0.49 | -0.64 | -0.54 | -0.20 |
| **$R^2$** | 0.24 | 0.41 | 0.29 | 0.04 |

**Tabel 1: Summary of the plume measurement results. Mean ambient $O_3$ levels at the ground and at the height of the plume are determined by averaging over the respective periods determined by the ambient pressure (and excluding data points with $SO_2$ mixing ratios larger than 0.5 ppb). The maximum $SO_2$ value marked with a * indicates that the sensor was in saturation and the real value is likely higher than reported. The mean $O_3$ inside the plume is obtained by averaging over periods for which xSO2 > 0.5 ppb. The mean $O_3$ depletion is determined by comparing the mean $O_3$ inside the plume with the mean ambient $O_3$ at height of the plume. The**
**maximum $O_3$ depletion is calculated using the minimum $O_3$ value inside the plume and comparing it to the ambient $O_3$ at height of the plume.**