# Peer review of "VOLCANO3 - A Miniaturized Chemiluminescence Ozone Monitor for Drone-Based Measurements in Volcanic Plumes"

_EGUsphere, 2025_

## Referee Comment (RC1)

This manuscript presents the VOLCANO3 instrument, a light-weight drone-deployable ozone monitor. Its chemiluminescence means of detection is designed to avoid cross-sensitivity to SO2 and therefore produce dependable measurements showing the depletion of ozone within volcanic plumes.

As a proof of its viability, the system is deployed to gather data within the plume of Mount Etna. The results shows decreased ozone concentrations within the plume.

This work is within the scope of the AMT journal. The creation of this system represents a significant advance in the capacity of researchers to investigate ozone depletions within volcanic plumes. While ozone depletions have been measured by other means, this is the first publication demonstrating a system for monitoring ozone that is practicable to deploy on a drone and resistant by design to interference in the signal from volcanic SO2.

The advantages that this system offers are evident in the example Etna results. There is q good level of detail to allow for replication and development by other researchers.

Overall I recommend this manuscript for publication in this journal with minor corrections.

**1. Introduction**

The introduction provides a reasonable background to the field.

I would suggest that the following are additionally addressed:

- Prior studies such as Rüdiger et al. (2018) and Karbach et al. (2022) have deployed drone-based in-plume measurements for other gases, notably SO2. Such drone based systems should be briefly referenced .
- So as to highlight the advantages of the VOLCANO3 being drone-deployed, the introduction should overview the settings for prior O3 measurements within plumes (i.e. aircraft based or ground-based and relying on grounding plumes)

**2. The principle of CL-O3 monitors**

This section describes, technically, the theory of chemiluminescence and reports the overall calculations that produce mixing ratio numbers as equation 1 and 2. This is useful and generally well described. The assignment of units to the parameters here needs to be consistent, with explicit units for pressure and temperature.

**3. A compact CL ozone monitor**

This section would be improved by a photograph of the system, in addition to the schematic shown in Figure 1.

There is a mismatch between the text and Figure 1 – there are references to labels A-D in the text but no such labels on the Figure.

The section on CL-Monitor Characterization is useful for replicability. Some minor comments regarding this:

- It would be helpful to know if such correction/calibration is required for each deployment.
- The parameter $a_{cal}$ should be defined immediately after its use.
- The O3 generators for calibration are external to the main devoice. The specific O3 generators used in this study could be replaced with alternative model. Therefore, the wording should clearly state that "In this study [these devices] were used".

- I assume the constant O3 periods discussed on line 180 are during calibration. This should be explicitly stated.
- As written the text in lines 190-193 describes only generating a step-up in O3. If a step-down was also tested, as implied on line 194, this should be explicitly noted.

**4. Field measurements**

This section describes a field campaign at Etna where the VOLCANO3 instrument was deployed and produced promising results.

Section 4.1 and Figure 4 demonstrate that VOLCANO3 can produce typical vertical O3 profiles.

Section 4.2.2. describes the instrumentation used in the campaign. VOLCANO3 is paired with "little-RAVEN" as described in Karbach et al. (2022) for various measurements including SO2. This is a critical element of the system, as without these volcanic plumes could not be identified in the signal. The weight of little-RAVEN should be given in this section, as it is useful for the reader to know the combined payload of the two instruments.

"little-RAVEN" has an SO2 saturation point of 16 ppm. This is a significant limitation and the current presentation at line 250 is too late. I suggest presenting this information within section 4.2.2..

Section 4.2.3. discusses the four flights of the campaign.

These flights are mapped on Figure 5. This map should clearly indicate the launch and return points for the flights. I also suggest adding arrows to the flight path so the reader can see the direction of flight.

Figure 6 shows clear anti-correlation of SO2 and O3 for one of the flight data sets. This is a very interesting result. Data for all flights are tabulated in Table 1. It is unclear how results where SO2 was above the saturation level were treated in the calculation of summary statistics, this should be made clear to the reader

**5. Future developments**

This section makes some reasonable suggestions as to how the VOLCANO3 system could be developed, particularly in terms of reducing weight.

I would like to see discussion here that relate to the 16 ppm saturation point for SO2 measurements. This is currently a significant limitation, as it prevents identification of the most dense parts of the plume where near total ozone loss may be expected. Could VOLCANO3 be paired with alternative SO2 monitors?

**6. Discussion and conclusion**

At line 286 "ambient measurements in Heidelberg" are mentioned, but these are not mentioned in the paper.

At line 289 the measurement accuracy is reported to be around 7% for 40 ppb O3.

The final sentence of this section appears to be incomplete.

**Other comments, mostly technical**

- Throughout: Some numerical values use commas rather than dots for decimal markers. These should be dots throughout
- Throughout: The format of minxing ratios (ppmv, ppbv vs. ppm, ppb) should be consistent throughout.
- Line 13: Add "tropospheric" before volcanic plumes
- Line 14. Suggest the statement "the underlying chemical mechanisms are still poorly understood" be changed. There exists now a reasonable theoretical understanding of the associated chemistry, albeit with some unknowns.
- Line 57: "in use since decades" change is "have been used for several decades".
- Line 77: change "assumption" to "result" or similar. This phenomenon has been repeatedly observed and can be described in stronger terms than an "assumption".
- Line 120: Change format of reference.
- Line 127: the presence of "(:" suggests some text or label is missing here.
- Line 150: change "in" to "at"
- Line 211: suggest "Geological evidence suggsts volcanic activity since 0.6 million years" changes to "Geological evidence suggests it has been active for approximately 0.6 million year". Alternatively this sentence could be removed entirely.
- Line 212-213: suggest removing "is undergoing significant morphological changes over time", and changing "currently hosting" to "currently has".
- Line 247: remove "basically"
- Line 301-302: change "prove" to "proof", change "those theoretical considerations" to "these model predictions".
- Line 418-419: Check URL format
- Line 480: Change "Tabel" to "Table"

**References**

Karbach, N., Bobrowski, N., & Hoffmann, T. (2022). Observing volcanoes with drones: studies of volcanic plume chemistry with ultralight sensor systems. *Scientific Reports*, *12*(1), 17890. https://doi.org/10.1038/s41598-022-21935-5

Rüdiger, J., Tirpitz, J.-L., De Moor, J. M., Bobrowski, N., Gutmann, A., Liuzzo, M., Ibarra, M., & Hoffmann, T. (2018). Implementation of electrochemical, optical and denuder-based sensors and sampling techniques on UAV for volcanic gas measurements: Examples from Masaya, Turrialba and Stromboli volcanoes. *Atmospheric Measurement Techniques*, *11*(4), 2441-2457.

---

## Author Comment (AC1)

**Detailed response to Luke Surl comments**

We would like to thank Luke Surl for his positive review and for his helpful comments and suggestions to improve the quality and clarity of our manuscript.

For reference the original comments are always included in regular font style with our response following in italic style.

**1. Introduction**

The introduction provides a reasonable background to the field. I would suggest that the following are additionally addressed:

• Prior studies such as Rüdiger et al. (2018) and Karbach et al. (2022) have deployed drone based in-plume measurements for other gases, notably SO2. Such drone based systems should be briefly referenced.

**Response:**

We believe that this would distract too much from the actual topic of presenting a new developed miniaturised CL-O3 monitor. It would only unnecessarily lengthen the introduction, and we therefore prefer not to make any changes to the text in this regard.

• So as to highlight the advantages of the VOLCANO3 being drone-deployed, the introduction should overview the settings for prior O3 measurements within plumes (i.e. aircraft based or ground-based and relying on grounding plumes)

**Response:**

In our opinion this topic is already well covered in our introduction, citing most of the previous volcanic plume studies for O3. Nevertheless, we added a small clarification in line 95 emphasizing the logistical challenges accessing volcanic plumes for O3 measurements in previous efforts.

So far the following O3 measurements within plumes have been referred to, including the overview article of Vance et al., 2010:

"Consequently,  $SO_2$  typically dominates UV absorption in volcanic plumes and prohibits an accurate quantification of the  $O_3$  UV absorption signal (Kleindienst et al., 1993; Leston et al., 2005; Williams et al., 2006). The correction of the data with simultaneously measured  $SO_2$  (Kelly et al., 2013) or the application of selective  $SO_2$  scrubbers (Surl et al., 2015; Vance et al., 2010), however, are difficult and – at best - introduce significant additional uncertainty. "

"Field studies (using CL as well as short-path UV absorption instruments) have shown varying degrees of  $O_3$  depletion across different volcanoes, in some cases up to 90%  $O_3$  loss compared to ambient levels were reported (e.g. at Mount St. Helens, USA, see Hobbs et al., 1982). In other cases, no  $O_3$  depletion was found (e.g. at Kilauea, Hawaii, USA, see Roberts, 2018) which was explained by low concentrations of halogens and is supported by measurements by Kern et al, 2018. "

"Measuring O3 levels in volcanic plumes is challenging and often relies on substantial logistical efforts such as aircrafts or requires specific meteorological or topographical conditions to access the plume with ground-based instruments. The aim of this study is to provide a technique for reliable O3 measurements in volcanic plumes. Building upon previous studies, this work focuses on employing gas-phase chemiluminescence (CL)-based O3 monitors for volcanic plume measurements (Hobbs et al., 1982; Vance et al., 2010; Carn et al., 2011)."

**2. The principle of CL-O3 monitors**

This section describes, technically, the theory of chemiluminescence and reports the overall calculations that produce mixing ratio numbers as equation 1 and 2. This is useful and generally well described. The assignment of units to the parameters here needs to be consistent, with explicit units for pressure and temperature.

**Response:**

We have carefully reviewed the section again and added the requested information in line 115/116

"p is the ambient pressure in Pa, T the ambient temperature in K,"

We didn't find any further disagreement

**3. A compact CL ozone monitor**

This section would be improved by a photograph of the system, in addition to the schematic shown in Figure 1.

**Response:**

We now added a photograph of the system as Figure 1 b.

There is a mismatch between the text and Figure 1 – there are references to labels A-D in the text but no such labels on the Figure.

**Response:**

Sorry indeed this might have read misleading. With the new added Figure 1b we adapted the text accordingly.

The section on CL-Monitor Characterization is useful for replicability. Some minor comments regarding this:

• It would be helpful to know if such correction/calibration is required for each deployment.

**Response:**

The monitor showed quite stable behaviour, however a regular calibration is advised. We added a sentence:

"Although VOLCANO3 demonstrated stable behaviour, it is advisable to perform a regular calibration check before each measurement campaign."

• The parameter acal should be defined immediately after its use.

**Response:**

You are right we shifted the sentence "acal is the calibration constant" two lines above.

• The O3 generators for calibration are external to the main devoice. The specific O3 generators used in this study could be replaced with alternative model. Therefore, the wording should clearly state that "In this study [these devices] were used".

**Response:**

We agree and change accordingly. In line 168/169 it reads now: "The CL  $O_3$  monitor is calibrated using an  $O_3$  generator, **in our study** primarily the Ozone Calibration Source Model 306 by 2B Technologies."

• I assume the constant O3 periods discussed on line 180 are during calibration. This should be explicitly stated.

**Response:**

We are a bit confused by this comment. Former line 180 is indeed in the calibration section describing how the measured  $O_3$  values during the calibration are gained. We copy here the lines the reviewer is referring to and don't see a point for further clarification.

"The CL  $O_3$  monitor is calibrated using an  $O_3$  generator, in our study primarily the Ozone Calibration Source Model 306 by 2B Technologies. It is a portable  $O_3$  generator and can provide  $O_3$  in the range of 0 to 1000 ppb. Additionally, the  $O_3$  generator ANYSCO type SYCOS KT- $O_3$ /SO2, which can provide 0 and 150 ppb of  $O_3$ , was used. To calibrate the monitor several calibration measurements with varying  $O_3$  mixing ratios in different sequences are made. For the periods of constant  $O_3$ , the converted signals are averaged and plotted against the sampled  $O_3$  mixing ratios as shown in Fig. 3. "

• As written the text in lines 190-193 describes only generating a step-up in O3. If a step-down was also tested, as implied on line 194, this should be explicitly noted.

**Response:**

Accordingly, to the suggestion of the reviewer we added few words in former line 192, now213ff:

"Once a stable mixing ratio is achieved, the hose can be swiftly connected and disconnected to and from the monitor, respectively."

**4. Field measurements**

This section describes a field campaign at Etna where the VOLCANO3 instrument was deployed and produced promising results.

Section 4.1 and Figure 4 demonstrate that VOLCANO3 can produce typical vertical O3 profiles. Section 4.2.2. describes the instrumentation used in the campaign. VOLCANO3 is paired with "little RAVEN" as described in Karbach et al. (2022) for various measurements including SO2. This is a critical element of the system, as without these volcanic plumes could not be identified in the signal. The weight of little-RAVEN should be given in this section, as it is useful for the reader to know the combined payload of the two instruments.

**Response:**

We added the weight and a little description of little RAVEN in the text "SO2/CO2 sensor "little-RAVEN" (Karbach et al., 2022) with 868 MHz radio link (RFDesign, approx. 3 km range), GPS module for time and position (MTK3339 Adafruit), Alphasense electrochemical SO2 sensor (calib. range: 0-16 ppm), CO2 sensor (K30 FR Senseair, not used in this work), temperature, humidity & pressure sensor (BME280). Total weight: approx. 300 g"

"little-RAVEN" has an SO2 saturation point of 16 ppm. This is a significant limitation and the current presentation at line 250 is too late. I suggest presenting this information within section 4.2.2. Section 4.2.3. discusses the four flights of the campaign.

**Response:**

The calibration range of little RAVEN is now mentioned in section 4.2.2 of the revised manuscript. Further the text is adapted in line 335-339 and the limiting factor of the SO2 sensor range is pointed out again as a future improvement for plume studies:

"For instance, to fully answer the question on  $O_3$  distributions in volcanic plumes and if  $O_3$  might be a limiting factor on the bromine transformation in volcanic plumes, more comprehensive measurement campaigns are essential and care should be taken to complement the  $O_3$  measurements by applying an  $SO_2$  sensor which covers the entire range of  $SO_2$  mixing ratios in the plume under investigation."

These flights are mapped on Figure 5. This map should clearly indicate the launch and return points for the flights. I also suggest adding arrows to the flight path so the reader can see the direction of flight.

**Response:**

We followed the suggestions and adapted Figure 5 as suggested.

Figure 6 shows clear anti-correlation of SO2 and O3 for one of the flight data sets. This is a very interesting result. Data for all flights are tabulated in Table 1. It is unclear how results where SO2 was above the saturation level were treated in the calculation of summary statistics, this should be made clear to the reader

**Response:**

Values where  $SO_2$  was above the saturation level or equal were not considered for the calculation of summary statistics. Formally this had been stated already in the Figure caption:

"In-plume datapoints are defined according to the SO2 mixing ratio for values larger than 1.5 ppm and smaller than the saturation value of 16 ppm."

However, we added this information also to the table caption:

"The mean  $O_3$  inside the plume is obtained by averaging over periods for which xSO2 > 1.5 ppmv and below 16 ppmv"

**5. Future developments**

This section makes some reasonable suggestions as to how the VOLCANO3 system could be developed, particularly in terms of reducing weight.

I would like to see discussion here that relate to the 16 ppm saturation point for SO2 measurements. This is currently a significant limitation, as it prevents identification of the most dense parts of the plume where near total ozone loss may be expected. Could VOLCANO3 be paired with alternative SO2 monitors?

**Response:**

Certainly, SO2 sensors with a larger range are commercially available and often applied and have been even used by part of the authors of this article in earlier works as referred to already by the reviewer himself - for instance Ruediger et al used an SO2 sensor up to 200 ppm. Our article is rather a proof of concept paper for the newly developed O3 monitor. But certainly, in future investigations of volcanic plumes SO2 sensors with a larger range should be used. As this is not part of a needed development we added this point to our discussion and conclusion section:

**Line 296-298**

"and care should be taken to complement the O3 measurements by applying an SO2 sensor which covers the entire range of SO2 mixing ratios in the plume under investigation"

**6. Discussion and conclusion**

At line 286 "ambient measurements in Heidelberg" are mentioned, but these are not mentioned in the paper.

**Response:**

We thank the reviewer for this notification. In an earlier draft of the manuscript we had also included vertical profile measurements from Heidelberg which were later excluded for easier comprehension and to avoid redundancy as we have included a vertical profile taken during the campaign at Mt Etna (Figure 4)

But overlooked this remaining half sentence noted by Luke Surl. The sentence has been changed now.

"Calibration measurements in Heidelberg, as well as measurements in the field, ..."

At line 289 the measurement accuracy is reported to be around 7% for 40 ppb O3. The final sentence of this section appears to be incomplete.

**Response:**

We don't find the incomplete sentence, mentioned by the reviewer.

Other comments, mostly technical

• Throughout: Some numerical values use commas rather than dots for decimal markers. These should be dots throughout

**Response:**

Thanks for noting that we revised accordingly.

• Throughout: The format of mixing ratios (ppmv, ppbv vs. ppm, ppb) should be consistent Throughout.

**Response:**

We revised the manuscript accordingly.

• Line 13: Add "tropospheric" before volcanic plumes

**Response:**

We don't agree here with the reviewer as O3 depletion can also take place in volcanic plumes located in the stratosphere.

• Line 14. Suggest the statement "the underlying chemical mechanisms are still poorly understood" be changed. There exists now a reasonable theoretical understanding of the associated chemistry, albeit with some unknowns.

**Response:**

We changed to: "The underlying chemical mechanisms are still incompletely understood"

• Line 57: "in use since decades" change is "have been used for several decades".

We revised the manuscript accordingly.

• Line 77: change "assumption" to "result" or similar. This phenomenon has been repeatedly observed and can be described in stronger terms than an "assumption".

**Response:**

We don't agree with the reviewer as most of the O3 measurements in volcanic plumes have been carried out without or very incomplete investigation of reactive halogens in volcanic plumes. Therefore, scientifically spoken it is still rather an assumption than a confirmed result.

• Line 120: Change format of reference.

**Response:**

The reference format has been adapted to match the style of the manuscript.

• Line 127: the presence of "(:" suggests some text or label is missing here.

**Response:**

*There is nothing missing it is the start of an enumeration : 1)*

• Line 150: change "in" to "at"

**Response:**

Changed as suggested.

• Line 211: suggest "Geological evidence suggests volcanic activity since 0.6 million years" changes to "Geological evidence suggests it has been active for approximately 0.6 million year". Alternatively, this sentence could be removed entirely.

**Response:**

Changed as suggested.

• Line 212-213: suggest removing "is undergoing significant morphological changes over time", and changing "currently hosting" to "currently has".

**Response:**

Done as suggested.

• Line 247: remove "basically"

**Response:**

Changed as suggested.

• Line 301-302: change "prove" to "proof", change "those theoretical considerations" to "these model predictions".

**Response:**

Changed as suggested.

• Line 418-419: Check URL format

**Response: Done.**

• Line 480: Change "Tabel" to "Table"

Response: Done.

---

## Author Comment (AC2)

**Detailed response to anonymous reviewer 2 comments**

We would like to thank the reviewer for their critical but constructive review and for the helpful comments and suggestions to improve the quality and clarity of our manuscript. We are confident that we were able to satisfy all questions and concerns.

For reference, the original comments are always included in regular font with our response following in italic font.

**General Comments**

This manuscript describes the development of an ethylene-chemiluminescence (ET-CL) ozone (O3) instrument for use on small drones (UAS), for the purpose of obtaining measurements from volcanic plumes. The authors provide a technical description of their instrument and results from an initial field test at Mt. Etna. Application of CL techniques to measure O3 in volcanic plume studies is an excellent idea since volcanic SO2 interferes with UV-absorption-based O3 instruments, and miniaturizing a CL instrument for drone use is novel. .. The manuscript needs significant revision to be appropriate for publication. Please see below for more detailed comments.

**Specific Comments**

A significant omission in the manuscript is that the well-known and dangerous nature of ethylene is never mentioned. This was a primary reason the U.S. EPA moved away from the technique, as described in Long et al., 2014 (pg. 5):

"3.1.3. Disadvantages The method requires a constant supply of ethylene, which is a dangerous, flammable, and potentially explosive gas typically stored in high-pressure gas cylinders. The use of such gas cylinders may be inconvenient and is often restricted by building fire codes or other monitoring site limitations."

Also in Spicer et et al., 2010: "The chemiluminescence method has been currently replaced in the United States by a Federal Equivalent Method (FEM), UV absorption (UV). A switch to the UV method occurred to reduce operational costs and improve safety by eliminating the flammable compressed ethylene gas required by the FRM."

The hazardous nature of ethylene should be discussed and recommendations for safe handling. As described, is the setup subject to hazardous materials rules (e.g. UN1950)? What limitations might the use of ethylene present for the practical application of this method? Were extra approvals needed to comply with aviation rules to carry ethylene on a UAS? If special steps were taken for permissions to operate a UAS carrying hazardous material, it would be useful to describe them to understand the practicality and potential broad applicability of the method.

**Response:**

We acknowledge the reviewer's safety concerns about handling ethylene in CL monitors. However, we also want to draw the attention to the fundamental differences between standardized and institutionalized air quality measurements and scientific field measurements at active volcanoes. Besides, the used gas amounts are far below limits for hazardous classification and the operation in the open atmosphere efficiently prevents accumulation of higher concentrations. We clarify these aspects in an added paragraph within the newly added Section 2.1:

**"2.1 Selection of chemiluminescence technique.**

The ethylene chemiluminescence (ET-CL) reaction was selected for ozone detection in UAV applications because it provides high photon yield in the visible range ( $\lambda \approx 440$  nm) and operates stably without active drying. In contrast, the nitric oxide chemiluminescence (NO-CL) reaction produces electronically excited NO2\*, emitting primarily in the red–near-infrared region with a broad maximum around 1200 nm (Clough, 1967). These long-wavelength photons have lower energy and are detected with markedly reduced quantum efficiency by standard photomultipliers, requiring cooled, red-sensitive detectors to suppress dark current noise. Such detector assemblies substantially increase mass and power consumption, which is critical in UAV applications.

The effect of water vapour also differs fundamentally between the two chemiluminescence systems. For NO-CL, water acts as an efficient collisional quencher of NO2\* emission, strongly reducing signal intensity and linearity (Matthews et al., 1977). For ET-CL, in contrast, water vapour slightly enhances the chemiluminescence signal through secondary excitation of formaldehyde and OH\* products, leading to a small positive bias rather than suppression (Kleindienst et al., 1993). Consequently, the humidity response of ET-CL can be accounted for in the measurement uncertainty, whereas NO-CL requires complete gas drying and thermal stabilisation to achieve reproducible sensitivity.

Considering these differences in spectral emission, humidity response, and detector requirements, the ET-CL configuration provides the most practical balance between analytical performance, stability under ambient conditions, and compatibility with compact, low-power UAV payload operation.

Although ethylene is flammable, only a very small amount of gas was used, contained in a sealed aluminum minicylinder. The instrument was usually operated in the open atmosphere. The gas volume was far below limits typically relevant for hazardous classification. Consequently, the ET-CL setup did not represent a relevant safety risk during field deployment."

• 28: re-word the sentence "Besides its prominent role and abundance in the stratosphere, smaller amounts of O3 in the troposphere play an important role in the oxidation chemistry." – perhaps "Besides its prominent role and abundance in the stratosphere, O3 is an important oxidant in the troposphere."

**Response:**

**Done as suggested**

• 32: "...measurements of the vertical profile with high spatial and temporal resolution are rare, yet highly desirable."

UAS offer an interesting new platform to potentially obtain O3 profiles, but it would be appropriate to mention existing global ozonesonde networks here, such as those organized by GAW/WMO, NDACC, NASA, NOAA, and SHADOZ (many of these efforts and related publications are summarized at https://tropo.gsfc.nasa.gov/shadoz/index.html). For example, Stauffer et al. (2022) summarized 42,042 sonde profiles from 60 global stations that tracked O3 from the surface to 30 km altitude from the years 2004-2021, demonstrating that considerable effort has gone into obtaining high-resolution, global O3 profile data. Perhaps "rare" should be qualified, or the type of vertical profile that is meant could be clarified. From a practical standpoint it is important to note that most countries restrict the altitudes at which UAS can operate without special permission, which presents a significant limitation for using

UAS to obtain vertical profiles. However, small UAS might have some advantages over other methods, such as tethered ballons, for low-altitude (i.e., boundary layer) studies.

**Response:**

We definitely agree with the reviewer that there are global ozone-sonde networks which have been operating for a long time already. However, we now notice that we were not very clear in our wording and our text might have been misinterpreted by the reviewer. Indeed, we mean sounding the  $O_3$ -profile in the lowest part of the atmosphere (up to around 1000m). We are sorry for this misunderstanding and thankful for the opportunity to clarify the sentence. We added the following text:

"measurements of the vertical profile with high spatial and temporal resolution in the lower troposphere (up to around 1000m), in particular the planetary boundary layer are rare."

• 35-37: "In fact, nitric oxide and ethylene CL measurements of O3 are still the standard method in the United States (USEPA, 2023) and are considered the most reliable O3 measurement methods (e.g. Long et al., 2014, Long et al 2021)."

This is partially true, but needs to be edited to clarify that the Ethylene-chemiluminescence (ET-CL) technique has been superseded by the Nitric Oxide-chemiluminescence (NO-CL) method, as summarized in Long, 2021:

"The ET-CL method is no longer used nor produced commercially and has been replaced by the NO-CL method...The ET-CL method was promulgated as the Federal Reference Method (FRM) for measuring O3 in the atmosphere in 1971, and the NO-CL method was promulgated as the FRM in 2015 (U.S. EPA, 2015)."

Further information on the EPA's rationale to move from ET-CL to NO-CL as a reference method is found on pages 65428-65429 in the U.S. EPA (2015):

"The existing O3 FRM specifies a measurement principle based on quantitative measurement of chemiluminescence from the reaction of ambient O3 with ethylene (ET–CL). Ozone analyzers based on this FRM principle were once widely deployed in monitoring networks, but now they are no longer used for routine O3 field monitoring... Although the existing O3 FRM is still a technically sound methodology, the lack of commercially available FRM O3 analyzers severely impedes the use of FRM analyzers...Therefore, the EPA proposed to establish a new FRM measurement technique for O3 based on NO-chemiluminescence (NO–CL) methodology. This new chemiluminescence technique is very similar to the existing ET–CL methodology with respect to operating principle, so the EPA proposed to incorporate it into the existing O3 FRM as a variation of the existing ET–CL methodology, coupled with the same existing FRM calibration procedure."

Also, I could not find "USEPA, 2023" in the references, nor could I independently find updated rules on O3 FRM after 2015. The U.S. EPA webpage states "In December 2020, EPA decided to retain the current ozone standards set in 2015" (https://www.epa.gov/ground-level-ozone-pollution/setting-and-reviewing-standards-control-ozone-pollution).

The information about ET-CL being superseded by NO-CL is elided in the present manuscript, and more background concerning the chosen measurement technique (ET-CL) should be included. I suggest reviewing these references, fixing the USEPA, 2023 reference

in the manuscript, and providing a more thorough background that explains why ET-CL was chosen for this application.

**Response:**

We thank the reviewer for this question, which again gives us the opportunity to clarify our point. The choice of the ethylene chemiluminescence (ET-CL) system was driven by the optical and operational constraints of UAV payload operation. While NO-CL can achieve very low detection limits in laboratory setups, the light it emits originates mainly from electronically excited  $NO_2^*$  in the red to near-infrared spectral ranges (peak  $\approx 1200$  nm; Clough, 1967). Its detection requires red-sensitive photomultipliers which need to be cooled in order to suppress dark current noise. These detectors would add substantial mass and power consumption. In addition, the NO-CL emission is efficiently quenched by water vapour, which introduces interference to water vapour and causes loss of linearity and sensitivity (Matthews et al., 1977). ET-CL, by contrast, emits at shorter wavelengths ( $\approx 440$  nm) and shows a very small positive humidity bias that can be quantified and included in the uncertainty. We are convinced that this trade-off provides the best balance between analytical precision, ambient stability, and UAV compatibility.

These considerations are also explained in the new subsection ("2.1 Selection of chemiluminescence technique", p. 4f l. 122-137) which we added to the revised manuscript. (See text above).

We are also sorry that we forgot the USEPA, 2023 reference. It is now included in the list of references. Here ET and NO CL monitors are equally named as reference measurement principle

US Environmental Protection Agency (EPA): C.F.R., Appendix D to Part 50, Title 40, Reference Measurement Principle and Calibration Procedure for the Measurement of Ozone in the Atmosphere (Chemiluminescence Method), https://www.ecfr.gov/current/title-40/part-50/appendix-Appendix D to Part 50 (last access: 8 Nov 2024), 2023.

• 52-54: "The correction of the data with simultaneously measured SO2 (Kelly et al., 2013) or the application of selective SO2 scrubbers (Surl et al., 2015; Vance et al., 2010), however, are difficult and – at best - introduce significant additional uncertainty."

I agree that using filters like those described in Surl et al., 2015 and Vance et al., 2010 for proximal plume measurements with high SO2 loadings is not ideal. However, it's worth pointing out that both Vance et al. (2010) and Kelly et al. (2013) reported airborne measurements from dilute plumes where such interferences could be considered minor, and both studies found significant O3 depletions in volcanic plumes that was much larger than any potential artifact:

In the case of Vance et al., several sets of measurements are included from a variety of techniques, but observations with the largest  $O_3$  deficits come from airborne intercepts of the aged Eyjafjallajökull plume where co-measured  $SO_2$  was less than 120 ppbv. This much  $SO_2$  would result in a maximum of ~1-2 ppbv positive interference in a UV  $O_3$  instrument which is negligible compared to 10's of ppbv of  $O_3$  loss relative to ambient levels (Vance et al., 2010, Supplement, Table 2).

Kelly et al., 2013 reported airborne intercepts of plumes from Redoubt Volcano, with  $SO_2$  peak levels reaching up to only 1.2 ppmv, and most levels were lower. Their discussion points out the strengths and weaknesses and errors associated with their approach: see Section 3.4, e.g.

"Use of an interference-free technique to measure O3, such as chemiluminescence, would be preferable for making observations in volcanic plumes but unfortunately is not always practical. We acknowledge that the method we describe below has significant uncertainty but it has the advantage of using more common, inexpensive, and portable O3 and SO2 sensors to obtain information about O3 in SO2-rich volcanic plumes..."

However, Table 3 shows that the correction for  $SO_2$  interference was generally quite small (a few ppbv  $O_3$  – see  $O_{3(raw)}$  vs.  $O_{3(correct)}$ ), and that in most cases even the uncorrected data showed lower in-plume  $O_3$  levels than ambient  $O_3$ . In other words, the observed in-plume  $O_3$  depletions in the presented measurements were generally larger than the artifacts introduced by  $SO_2$  interference.

Thus, diminishing these previous studies does not seem justified. Instead, it could be noted that such approaches are best carefully applied in dilute plumes, and that another approach (ie, chemiluminescence, like that described in the present study) is generally advantageous and could be considered necessary for measuring O3 in dense and/or young volcanic plumes.

**Response:**

It was never our intention to diminish the work of other authors and, and we fully agree that in very diluted plumes with  $SO_2$  in the ppb range interferences are small or negligible. In our introdution we also specified that:

"Under most atmospheric conditions these interferences (especially due to  $SO_2$ ) are negligible (Kleindienst et al., 1993; Williams et al., 2006) since ambient  $SO_2$  levels are typically comparable to or lower than  $O_3$  levels.

Further we wrote: However, when probing volcanic emissions, SO2 mixing ratios may reach values up to several ten ppmv.."

However, to make this even more clear we added half a sentence in line 95 of our introduction: " in particular during the first hour after the emission"

While we agree that there are cases where the SO2 interference might be small and could be corrected we note that this is usually not known beforehand and bringing an isntrument to the field that may or may not work under the given circumstances ist not very useful. Therefore, we do not understand why the reviewer argues so strongly against the traditional ET-CL solution, as the reviewer confirms that this technique has been applied as a standard technique.

The strongest argument to not use ET-CL anymore as a reference technique is that this instrument is no longer commercially available. However, this does not speak against the actual technology, and there is no doubt that this technology was the standard technology in the US for many years.

We therefore don't see a necessity to change the sentence in lines 52-54 (numbering in the original manuscript).

• 60: As written, it's not clear what 'traditional' means here. This should be clarified with the expanded background that distinguishes between the CL-ET and CL-NO approaches.

**Response:**

Regarding the therm "traditional" we think we made the meaning of the term clear. Also, in the new section 2.1 (see above) we expanded the background and distinguish between CL-Ethylene and CL-NO as requested.

• 82-83: Kern et al., 2020 should be added here, as they found very low BrO/SO2 ratios in the 2018 eruptive plume from Kilauea (see page 55), which corroborates with the low ozone depletion reported by Roberts (2018).

**Response:**

We thank the reviewer for this suggestion and added the above reference as suggested. Changed text:

"... was explained by low concentration of halogens and is also supported by measurements of Kern et al., 2020"

79-88: I'm not sure if I understand the model, as described, that predicts minimal ozone destruction in volcanic plumes (and the URL link in the reference did not work for me). Does this model assume constant influx of O3 into the plume? If so, it will not realistically capture the entrainment process during plume expansion. Furthermore, many "model studies with more evolved multiphase atmospheric chemistry mechanisms predict significant destruction of O3 in volcanic plumes", for example many well-known works by von Glasow (e.g. 2003, 2009, 2010), Bobrowski et al., 2007, Roberts (2009, 2014), and more recent works by their collaborators. While O3 measurements in volcanic plumes remain rare - and many unknowns remain about inplume halogen chemistry - most field measurements cited in the manuscript find significant O3 destruction (e.g. Hobbs et al., 1982; Vance et al., 2010; Carn et al., 2011, Kelly et al., 2013, Surl et al., 2015, etc.). Thus, despite a paucity of in-plume O3 measurements, existing field studies tend to agree in general with published plume chemistry models that include multi-phase reactive halogen chemistry that is kickstarted by mixing of hot halogen-rich volcanic gases with ambient air. The purpose of this section should be clarified, and adequate information and supporting references are needed if the intended purpose is to draw distinctions between models that predict or don't predict ozone depletion in volcanic plumes.

**Response:**

Our point is that a simple model assuming typical figures for turbulent diffusion ( $K=10^5$  cm2/s, see e.g. Brasseur & Solomon ...) in the free atmosphere will predict no significant O3 depletion. Models making other assumptions (e.g. Glasow et al. 2003, 2009, 2010, Bobrowski et al., 2007, Roberts et al. 2009, 2014...) find  $O_3$  depletion, however it is unclear whether the assumptions made in these models are realistic. Furthermore, when measurements of  $O_3$  depletion in volcanic plumes were made, in most cases the halogen loading of the plume was

not measured. Therefore, in our opinion it is important to verify the model predictions by more measurements in the field together with halogen measurements in the future.

• 100: "...with C2H4 being the most commonly employed reactant and which is used also in this study..."

Again, this is no longer the case and C2H4 has been superseded by NO (e.g. U.S. EPA 2015, Long et al., 2021). Please amend here and throughout. The choice to build an ET-CL instrument rather than an ET-NO instrument needs to be explained.

**Response:**

We changed the quoted sentence and added - as described above - a new section 2.1 to motivate our choice for  $C_2H_4$ .

The new sentence read as follows: "In this study we used  $C_2H_4$ , which we motivate below in section 2.1"

• "The principle of CL O3-Monitors" section: in addition to the theoretical description, this section should include practical information concerning the strengths and weaknesses of the technique. In addition to the problematic (flammable and explosive) nature of ethylene identified earlier, well-known positive interferences from water vapor in this type of instrument are not mentioned. This water vapor interference (which quenches the ET-O3 reaction) is described in several references (e.g. U.S. EPA, 2015), including two already referenced by the authors (Kleindienst, 1993; Long et al., 2021). Example descriptions of the sense and magnitude of the water vapor interference are found in Kleindienst, 1993:

"The chemiluminescence-based monitors showed systematically higher readings than the UV monitors with added water vapor. The effect was found to be linear with water vapor concentration with an average positive deviation of 3.0 percent per percent H2O at 25 degrees C. For these measurement, ozone concentrations ranged from 85 to 320 ppbv and water concentrations from 1 to 3 percent (i.e., dew point temperatures from 9 to 24 degrees C). These results are largely in agreement with previous studies conducted to measure this interference, although the present study extends the range of water concentrations tested."

A more recent summary is given by Spicer et al., 2010:

"Historically two methods have been widely used for ambient air O3 monitoring. The ethylene chemiluminescence Federal Reference Method (FRM) was dominant in the United States during the 1970s and 1980s. The only common documented interference to this method is water vapor. The extent of the positive bias is on the order of 3–4% of the O3 reading for each percent (10,000 parts per million [ppm]) of water vapor in the air."

Note: Spicer et al. (2010) independently found positive interference of 3 to 10 ppbv O3 per 10,000 ppmv H2O at O3 levels from 55-200 ppbv (3-9%) for a commercially available ethylene CL analyzer (Table 4).

Finally, the U.S. EPA commented on water vapor interference in ET-CL instruments when considering its rule change (U.S. EPA, 2015, pg. 65429):

"2. Comments on the FRM for O3

Comments that were received from the public on the proposed new O3 FRM technique are addressed in this section. Most commenters expressed general support for the proposed changes, although a few commenters expressed some concerns. The most significant issue discussed in comments was the relatively small but nevertheless potentially significant interference of water vapor observed in the ET–CL technique...However, in further response to these commenters' concerns, the EPA has modified Table B–3 to extend this water vapor mixing requirement to newly designated ET–CL analyzers, as well. These measures should insure that potential water vapor interference is minimized in all newly designated FRM analyzers."

Why is water vapor interference not included in the testing or error budget? In addition to describing the problem, simple solutions to the interference using simple commercially-available dryers are tested and analyzed (e.g. Long et al., 2021). According to a text search, the only mention of "water" in the manuscript is 1.70, where it is listed as the first (and presumably most abundant) component of primary volcanic gas emissions. This is potentially very important given how water-rich volcanic plumes are. Plumes routinely contain 1000's to 10,000's ppmv of water vapor (especially close to the source), which suggests that the reported ET-CL measurements could have potentially significant positive artifacts, unless this issue was mitigated somehow. Please describe any testing or mitigation tactics for dealing with water vapor during the development and field testing, how artifacts are dealt with in the results, and please clarify if water vapor was measured as part of the sensing package.

**Response:**

We appreciate the detailed references and suggestions, many of the named references are already cited in our original manuscript and therefore known to the authors. As mentioned above we have added an additional section regarding the choice of ET containing also a rough estimate on the water vapor influence (new section 2.1, see above).

We discussed the flammable (and potentially explosive) nature of ethylene before (see above). We also noted that a small, positive water vapour interference exists, (which is not due to quenching the excited product of the ethylene+ $O_3$  reaction, since this would lead to negative water vapour interference). Under typical ambient conditions with 1%  $H_2O$  in the atmosphere this interference will amount to an additional  $O_3$  signal of 3-4 ppb and less at higher altitudes where the low temperatures prevent such high  $H_2O$  levels. We also note that — while water vapour is the main constituent of most volcanic emissions — in a cold plume, where all these measurements are made the water vapour can not exceed saturation levels. Furthermore, the water vapour interference of the ET-CL method is an intrinsic feature of the method and there is no need to test it again and again.

• 124: Please include a photograph of the instrument to accompany the schematic shown in Fig. 1. For example, Figure 13 from Bräutigam, 2022 might be appropriate.

**Response:**

We included now a photograph from the instrument, see new Figure 1b

• 153: If I understand Figure 2 correctly, the dark current appears to vary from ~4-12 ppbv equivalent O3 at temperatures from 15-35°C. How is the derived uncertainty of the fit so small ("1 ppb")?

Response: The reviewer understood correctly that the dark current varies from  $\sim$ 4-12 ppbv equivalent  $O_3$  at temperatures from 15-35°C, but have probably overlooked our description in the text fort he correction of the dark current which is done depending on the recorded temperature Equation 3, line 76 – 80. So 1 ppb is the uncertainty of this correction.

• 171: "To calibrate the monitor several calibration measurements with varying O3 mixing ratios in different sequences are made." – this is too vague. Please elaborate on the calibration conditions and procedures. Also indicate if the calibration procedure utilized dried or humid air.

**Response:**

We agree that the original description of the calibration procedure was too brief in this regard and have expanded this in the revised manuscript. The calibration description in the revised manuscript now reads as follows:

"The CL  $O_3$  monitor is calibrated using an  $O_3$  generator, in our study primarily the Ozone Calibration Source Model 306 by 2B Technologies. It is a portable  $O_3$  generator and can provide  $O_3$  in the range of 0 to 1000 ppbv. Additionally, the  $O_3$  generator ANYSCO type SYCOS KT- $O_3$ /SO2, which can provide 0 and 150 ppbv of  $O_3$ , was used. In both instruments, ambient air is used as the feed gas and ozone is generated photolytically by UV irradiation of oxygen-containing air. Before entering the photolysis chamber, the incoming air is cleaned, and in the case of the  $O_3$  generator ANYSCO type SYCOS KT- $O_3$ /SO2 it is additionally dried by passage through a silica-gel drying tube.

For calibration the CL  $O_3$  monitor is directly connected to the O3 generator using a hose. We set the ozone mixing ratio at discrete steps (e.g. 0, ~10, ~50, ~100, ~150 ppbv) and held each step for several minutes to ensure sufficient statistical averaging. For each of these steps of constant  $O_3$ , the mean and the standard deviation of converted signals are calculated and plotted against the sampled  $O_3$  mixing ratios as shown in Errore. L'origine riferimento non è stata trovata. A linear fit is performed; its fit parameters are then used for the calculation of the  $O_3$  concentrations. Although VOLCANO3 demonstrated stable behaviour, it is advisable to perform a regular calibration check before each measurement campaign."

• 175: Change to "The detection limit..."

**Response:**

*Unfortunately, it is not clear to us what we should change here.*

• 180: here and throughout, make sure to include all equation variables and units, and be consistent (e.g. "ppm" appears 9 times in the manuscript, "ppmv" twice; "ppb" 20 times, ppbv once).

**Response:**

We read carefully through our manuscript again and made changes accordingly.

• 200: Was anything done to test and validate the instrument's performance at different elevations? Was it compared to a reference instrument? If so, please include and explain how this was done. What was done to assure the instrument would work well in plume conditions (ie, high elevation, humid)?

**Response:**

We thank the reviewer for this suggestion and agree that it would be beneficial to validate our method with reference technology. However, as we clearly pointed out in our manuscript, there is no reference technology for  $O_3$  measurements in volcanic plumes.

• 200: Was a vertical profile flown to evaluate atmospheric structure at altitudes relevant to the plume?

**Response:**

• We agree that such and similar measurements would be interesting. However, detailed studies of the small-scale phisicochemical structure of the atmosphere are beyond the scope of this technical manuscript. 221: please include more information on the "little-RAVEN". What sensors were included? What were the ranges/resolution/etc. What was the weight? Was it flown simultaneously with the VOLCANO3?

**Response:**

A full description of the little-RAVEN system is given in Karbach et al., 2022, however we understand, that a short description of the most important components in the manuscript might help the reader.

Therefore, we revised the following part to the manuscript:

- SO2/CO2 sensor "little-RAVEN" (Karbach et al., 2022) with 868 MHz radio link (RFDesign, approx. 3 km range), GPS module for time and position (MTK3339 Adafruit), Alphasense electrochemical SO2 sensor (calib. range: 0-16 ppm), CO2 sensor (K30 FR Senseair, not used in this work), temperature, humidity & pressure sensor (BME280). Total weight: approx. 300 g.
- Drone "Matrice 300 RTK", DJI, https://enterprise.dji.com/de/matrice-300/specs

The little-RAVEN sensor system, designed around an ESP microcontroller (ESP32 from Espressif), manages various sensors to determine SO2, CO2, temperature, humidity, pressure, and GPS location. It logs data onto internal memory and also transmits it to a ground station, allowing real-time localisation of the plume and confirming plume gas measurements. For this work, Little-RAVEN was used as a stand-alone system which could be attached to the measurement drone to provide the aforementioned information simultaneously with the ozone measurements.

- 230: Was the atmosphere stratified or compositionally heterogeneous with respect to the point of volcanic gas emission vs. the point of measurement in the buoyant plume? What was the difference in elevation from the vent to the altitude of the plume measurements? The manuscript only considers chemical ozone destruction and does not consider the impacts of mixing, entrainment, and transport of air from chemically dissimilar air parcels on measured O3 (as described by Kelly et al. 2013).
  - Were any BrO measurements made coincident with the O3 measurements?
    These would help to link O3 depletion to BrO formation.

**Response:**

These are interesting questions, but they are all beyond the scope of this manuscript. The manuscript was prepared as an article for AMT with the scope to introduce newly developed instruments which will make such studies in future possible. Therefore, the focus of the article is the description of the instrument and to show the feasibility of such investigations which we do by presented first experiments carried out in a volcanic plume.

• 230-253: the results section is weak. Ozone depletion has been measured many times (relatively speaking) at Mt. Etna. How do these new results compare to previous measurements and models (e.g. Vance et al., 2010, Roberts et al., 2014, Surl et al., 2015)? Did the CL technique obtain different results than these previous studies?

The result of this paper is the proof that the newly developed O3 monitor is giving reasonable results. This is a paper for AMT and not for ACP and as mentioned in the response to the question just above, so we don't agree with the reviewer that our result section is weak.

However, as mentioned below for another question of the reviewer we included the following words into the new version of our manuscript in line 334-336: "While our finding is consistent with measurements by Surl et al., 2015 (15% - 45%) and Vance et al., 2010 (15% - 40%),

249: The SO2 sensor range was only 16 ppm? Why? This seems like a serious limitation of the setup.

**Response:**

We chose a SO2 sensor in the lower concentration range by design to maximize precision in the range that we initially targeted (this is within the ranges mentioned in Vance et al. (2010) and Kelly et al. (2013) for airborne plume measurements). A lower concentration sensor provides higher gain, lower noise, and finer resolution at 0–15 ppmv than that of an e.g. 0–200 ppmv sensor, which has a higher noise floor/lower sensitivity at low concentrations. Our sampling and measurement strategy targeted traverses farther away from the source that kept in-plume values mostly within this range. While the low concentration range is certainly a limitation, we chose this sensor nevertheless to optimize performance at the targeted range. We monitored for any approach to the upper limit; areas above 16 ppmv were excluded from quantitative analysis, so saturation does not affect reported results.

**• 255: Improvements:**

As noted above, C2H4 is flammable and problematic from a hazardous materials standpoint. Is cyclohexane (C6H12) also problematic from a safety/hazardous materials standpoint? What about other methods such as NO-CL or so-called 'scrubberless' methods where N2O is converted to NO and used to titrate O3 (described in Long et al., 2021)? Also, what would need to be done differently next time to better characterize the plume and its chemistry?

**Response:**

An evaluation of NO-CL is now given in the added section 2.1 to the new version of our manuscript, please see above to similar questions

• L294: "Today, our knowledge is mainly based on model studies. Such model studies show for instance a complete ozone depletion in the centre of halogen rich volcanic plume after a relatively short distance from the emission point (about 10 min

downwind, Roberts et al., 2014) but a solid experimental prove of those theoretical consideration is still missing."

I disagree with this conclusion and suggest revisiting Roberts et al., 2014. They present several scenarios, some of which result in modest ozone depletion (see their figure 8).

**Response:**

Yes, we don't doubt that there are modest  $O_3$  depletion possible. But what is not proven today if we can have a complete ozone destruction in the centre part of the plume. Such proof can be only gained by airborne measurements.

Further in the article Roberts et al, 2014 the authors write in the text: "the single-box simulations presented here that predict the downwind trend do not simulate the ozone distribution across the plume cross section. Ozone loss is typically greater in the plume centre than near the edges"

Means the lines shown in Figure 8 are not spatially dissolved and don't exclude a heterogeneous distribution of  $O_3$  in the plume.

The author further argue that  $O_3$  could be a limiting factor for the formation of BrO, this still has to be proven, because this would be the case when  $O_3$  would be reduced close to 0 in part of the plume. Spatially resolved measurements in the plume were difficult before the advent of drone based measurement, so they are becomin now possible with the instrument developed in this study.

It's true that only one study (thus far that I'm aware of) has coupled airborne measurements with model results (Kelly et al., 2013). In that case the model parameterization was constrained as best as possible by field measurements and did not predict complete ozone destruction in the plume. In fact, good overall measurement-model agreement was found, suggesting that the model captured the major pieces of the chemistry in that case. Unfortunately many of the most important species involved in the relevant chemistry are hard to measure, so models are critical for understanding these unique systematics. This comment should be reconsidered, although I agree that more measurements are needed.

**Response:**

We are not sure how we should understand this statement of the reviewer. Since we already state that measurements are needed we saw no necessity for further changes in the text.

L.303: Given the flammable and explosive nature of ethylene, is it reasonable to suggest flying such a payload over fires or cities? The conclusions need to reflect the limitations of the current approach.

**Response:**

See above - additional section 2.1

**Table 1:**

• Can a plume age be calculated for the listed measurements?

**Response:**

An estimate of the plume edge would have a very high error as plume speed was not measured during the campaign. So unfortunately we cannot calculate a meaningful plume age

• Why is the correlation coefficient and r2 value listed? Isn't this redundant?

**Response:**

The reviewer is right but we would prefer to leave both values, although certainly  $R^2$  is not strictly necessary

• How do the derived O3/SO2 ratios compare to other measurements from Etna (e.g. Surl et al. 2015)?

**Response:**

The  $O_3$  depletion in Surl et al., 2015 (15-45%) and the one in Vance et al. 2010 (15-40%) are relatively similar to the one in our study were we determined a max.  $O_3$  depletion of 15-59% during the various flights.

The  $O_3/SO_2$  ratios are higher in our study as we measured in a more diluted plume compared to Surl at al. 2015

We adapted the text of the revised manuscript in line 334-336: "While our finding is consistent with measurements by Surl et al., 2015 (15% - 45%) and Vance et al., 2010 (15% - 40%), which also indicate a strong anti-correlation between SO2 and O3, the observed O3 depletion within the volcanic plume presents still an intriguing scientific puzzle."

• What was the humidity inside the plume?

**Response:**

We revisted our data aquisition, and as expected the RH was highly variable between the days and the flights. The highest relative humidity was measured on June 13th and resulted up to 90 %. In all other flights the relative humidity was significantly lower, down to 10% The measurements were carried out at altitudes mostly above 3300 m in the volcanic plume, so at relatively low temperature, means a high water vapor content cannot be expected and as explained earlier would result in a very small additional error so not change significantly our results. Upon acceptance of the manuscript all measurement data including RH measurements will be available on Zenodo and in accordance with AMT data availability policies.

• At which altitudes was the plume measured?

**Response:**

The pressure displayed in color on the  $O_3$  measurement plots is a good indicator for the elevation. The plume height was close to the summit crater height which is about 3300 m, we don't think that this is a relevant information for the table.

**Figure 2:**

• What are 'normal' measurements? Please clarify or refer the reader to the text where this is described.

**Response:**

We agree with the reviewer that 'normal' measurements is not a meaningful phrase and therefore rephrased the Figure caption:

"The black points display dark current measurements taken in the field by using an  $O_3$  scrubber placed on the entrance, this was usually done before and after each measurement flight."

---

## Author Comment (AC3)

**Detailed response to anonymous reviewer 3 comments**

We would like to thank the reviewer for her/his positive review and for his helpful comments and suggestions to improve the quality and clarity of our manuscript.

For reference the original comments are always included in regular font style with our response following in italic style.

**Major Comments:**

- There is no discussion of the effects of water vapour on the system, despite the listing of water vapour as the primary gas emission from volcanoes. This needs to be addressed, as there are % level biases reported for ethylene-CL with water vapour

Several authors (e.g. Kleindienst et al. 1993, Ollison et al. 2013, Spicer et al. 2012) agree that a positive interference to water vapour around 3-4 ppb O3 per 10,000 ppm of water vapour exists in ethylene CL instruments.

**Response:**

This appears to be an intrinsic feature of the method, so there is no need to measure it again with each implementation of an ethylene CL instrument. The revised manuscript includes a new section 2.1 which discusses the choice of the CL technique in this work and also includes a paragraph discussing the water vapour interference.

"Measurements with the VOLCANO3 instrument are typically performed in volcanic plumes that have cooled to ambient atmospheric temperature, so that atmospheric water vapour concentration cannot exceed local saturation level. Typical  $H_2O$  concentrations range from 1000 ppmv – 20000 ppmv and are lower at high altitudes and cold atmospheric temperatures. This adds an uncertainty of  $O_3$  measurements of about  $\pm 4$ ppbv when assuming 10,000 ppmv water vapour as "standard". Note that even though water vapour is the most abundant component of volcanic emissions, its mixing ratio cannot exceed saturation level.

We also added the term to the uncertainty calculation in section 3.2.3:

"The fifth term in Eq. (7) accounts for the well-established positive interference of water vapour on ethylene chemiluminescence. This interference is approximately 3–4 ppbv  $O_3$  per 10,000 ppmv  $H_2O$  (Kleindienst et al., 1993; Spicer et al., 2012; Ollison et al., 2013). As volcanic plume measurements are conducted at ambient atmospheric conditions, water vapour cannot exceed local saturation and typically ranges between 1,000 and 20,000 ppmv.

To include this effect in the instrument uncertainty, we assume a typical value of 10,000 ppmv and thus treat the humidity bias as a constant additive systematic uncertainty,  $\Delta H_2O$ , given by:

$$\Delta_{\mathrm{H_2O}} pprox 4~\mathrm{ppbv}$$
 . "

- Section 3.2.1 describes how the PMT's dark current is corrected for based on a fit with cell temperature rather than maintaining a constant temperature. L150 states that the fit parameters "are determined at regular intervals" which is very vague. It is not clear to me if this is something that has been determined in the laboratory, or whether the instrument is capable of determining this

during normal operation. If this is the latter, I don't see from the instrument description or figure 1 how this would be achieved.

**Response:**

The shown dark current measurements in Figure 2 are measurements partly taken during laboratory experiments but also measurements taken in the field. Usually we took dark current measurements before and after each flight by simply placing an O3 scrubber at the inlet. To this data the Richardson function is fitted, which is then used to determine the dark current for other measurements. In other words, the dark current correction is determined in the laboratory (and crosschecked with dark current datapoints from measurements not specifically done to determine the dark current) and then only applied to the measurements. So the fit is done only once.

We agree with the reviewer that the original text might sound somewhat unclear. We therefore rephrased it to better explain our entire procedure of the dark current correction:

"In order to subtract the dark current as a function of temperature from the data the Richardson function (see Eqn. 3) is fitted to the dark current data points (see Fig. 2):

The values of the fit are  $a_1 = (9.9 \pm 0.3) \cdot 10^{12} \text{ mv/K}^2$ ,  $b_1 = (-1.2 \pm 0.001) \cdot 10^4 \text{ K}$ , and  $c_1 = (0.55 \pm 0.003) \text{ mV}$ .

- Figure 3 shows the measured dark current, and within the relevant temperature range there is variation of  $\pm$  5 ppbv of O3 from the fit. It needs to be more clearly demonstrated how the authors arrive at only 1 ppb of additional uncertainty from this.

**Response:**

As explained in section 3.2.1 we performed a dark current correction based on the measured temperature of the instrument. In this section we also – in our opinion clearly – explain how the error of the dark current correction propagates into the error of the final result.

- More broadly a better diagram of the instrument would substantially improve the readers understanding. For example, the authors note that the ethylene flow is controlled via capillary, but do not state how flow through the instrument is controlled or measured – though this is required for their correction factor Ccon. Without sufficient control would this not vary with altitude?

**Response:**

The measurement of the ethylene flow through the instrument is calculated from the pressure measured at the minican and the ambient pressure since the flow through a capillary is determined by the pressure difference at its ends. The relationship between the pressure difference at both ends of the capillary and the flow through the capillary was determined experimentally with specific experiments in the lab, where we used pneumatic trough measurements. As a sanity check we also used the volume loss in the minican during measurement to determine the mean flow of ethylene during the measurement. Both experiments are in good agreement and can be empirically fitted. This fit can then be used to correct the change in ethylene flowduring the measurements. The variations arise mainly from the decreasing fill level of the minican but also due to altitude as the reviewer correctly noted, and are accounted for by continuously measuring both the minican pressure and the ambient pressure.

The rate of the total flow (air plus ethylene)  $f_{tot}$  through the cell is controlled by the rotary vane pump. A step-down converter enables to change the supply voltage continuously between 0:5 - 4:8 V (for an input voltage of 5 V), changing the flow rate from 5 to 34 mL/s. During the measurement the pump voltage was set to a flow rate of 21 mL/s.

As the reviewer correctly notes the calibration depends on the temperature and pressure at which the instrument operates, this dependence is described in Eq. (2).

- No mention of the time resolution (data acquisition time or averaging intervals) of the instrument is mentioned. From figure 6 data is presented at what appears to be ~2 Hz? This is relevant broadly for the reader to understand the instrument's capabilities, but also for the context of performance statistics such as LOD. LODs reduce with averaging, and the reported LOD of 1.13 ppb appears low for a 2 Hz measurement, with the uncertainty from dark current alone presented as 1 ppb. Moreover the stated LOD assumes the standard deviation in the dark current to be 0.4 mV, which from figure 3 appears to be a favourable case? Do the authors have data that they could use to perform Allan variance to provide further information on the LOD?

**Response:**

We thank the reviewer for this insightful comment. The instrument is sampling rate is roughly 3.5 Hz, and all uncertainty and detection limit calculations in Section 5.4 are based on the unaveraged time series. For the final plots (e.g. Fig. 6), we applied a 6 s rolling mean (20 data points are averaged) to improve readability. For the 6 s averaged data shown in the figures, the noise and thus the effective LOD are reduced.

- Section 3.2.4 Instrument response time describes the experiment as "swiftly connecting the hose to the monitor". More details are required on how this was conducted – was the process automated using a fast acting valve? If this was performed manually, I would expect this response time to be biased high. A good understanding of the time response is relevant to the previous point surrounding data acquisition and averaging, and should be put into context with how those data are presented.

**Response:**

The connection and disconnection of the hose were performed manually. We acknowledge that this could slightly bias the estimated response time toward higher values. However, since the response time was obtained from exponential fits to the monitor's signal ( $R^2 > 0.9$ ), a small delay during manual switching would mainly shift the response curve horizontally rather than affect its exponential shape. Therefore, the derived value represents a slightly conservative estimate of the true instrument response time. We have clarified this point in Section 3.2.4 and discussed the potential influence of manual switching in the text.

**Technical Comments:**

- Reference need to be checked, for example: L36 USEPA 2023 does not appear in the reference list, the DOI for Kleindienst 1993 returns "not found", the 2B Technologies manual is not referenced in the text.

US Environmental Protection Agency (EPA): C.F.R., Appendix D to Part 50, Title 40, Reference Measurement Principle and Calibration Procedure for the Measurement of Ozone in the

Atmosphere (Chemiluminescence Method), https://www.ecfr.gov/current/title-40/part-50/appendix-Appendix D to Part 50 (last access: 8 Nov 2024), 2023.

DOI for Kleindienst et al. 1993 corrected (DOI was correct, just the website apparently disappeared). We added the reference to the 2B Technologies manual. We replace the reference to: ,L36 USEPA 2023' by ,EPA 2020':

EPA 2020 - Integrated Science Assessment (ISA) for Ozone and Related Photochemical Oxidants (Final Report, Apr 2020), EPA/600/R-20/012, see: https://cfpub.epa.gov/ncea/isa/recordisplay.cfm?deid=348522

- Figures in the SI are referenced out of order to that in which they are presented e.g S4 is referenced on L88, S1 is not referenced until L234.

Thanks the reviewer for noting this error. We corrected the order of the Figures accordingly.

- Data availability – I see no reason why this data cannot be archived (along with processing code?) and referenced in this section, as per the AMT data policy?

Response: We agree with the reviewer and upon acceptance of the manuscript the data will be available on Zenodo.

L37 "nevertheless, nowadays and since many decades" needs rewording

*Response:* We reworded the sentence.

- L127 includes text from the figure caption for Figure 1.

*Response:* We deleted the additional text.

- L128 / 130 / 131 – the labels that this text is referring to do not appear on the figure, and instead seem to refer to Figure 13 of Bräutigam 2022.

Response: We added a new Figure 1b (a photo of the instrument as requested by other reviewers) and this includes also the before missing labels

- L128 t(h)rough

Response: Corrected

- L 211 "0.6 million years" should this read "0.6 million years ago"?

Response: No we didn't mean "0.6 million years ago" - the activity is still ongoing since then, therefore the text is not changed.

- L233 "Fig Figure 5"

Response: Done

- L266 The chemspider reference does not appear in the reference list, and I suspect provides a reference to the data where this is determined?

 $Response: We \ added \ the \ URL \ to \ chemspider \ (https://www.chemspider.com) \ to \ the \ reference \ list.$